# A single-cell transcriptomic atlas of the adult *Drosophila* ventral nerve cord

**Aaron M Allen†\*, Megan C Neville†, Sebastian Birtles, Vincent Croset, Christoph Daniel Treiber, Scott Waddell, Stephen F Goodwin\***

Centre for Neural Circuits and Behaviour, University of Oxford, Oxford, United Kingdom

**Abstract** The *Drosophila* ventral nerve cord (VNC) receives and processes descending signals from the brain to produce a variety of coordinated locomotor outputs. It also integrates sensory information from the periphery and sends ascending signals to the brain. We used single-cell transcriptomics to generate an unbiased classification of cellular diversity in the VNC of five-day old adult flies. We produced an atlas of 26,000 high-quality cells, representing more than 100 transcriptionally distinct cell types. The predominant gene signatures defining neuronal cell types reflect shared developmental histories based on the neuroblast from which cells were derived, as well as their birth order. The relative position of cells along the anterior-posterior axis could also be assigned using adult Hox gene expression. This single-cell transcriptional atlas of the adult fly VNC will be a valuable resource for future studies of neurodevelopment and behavior.

**\*For correspondence:**
aaron.allen@dpag.ox.ac.uk (AMA);
stephen.goodwin@dpag.ox.ac.uk (SFG)

†These authors contributed equally to this work

**Competing interests:** The authors declare that no competing interests exist.

## Introduction

The adult *Drosophila* central nervous system (CNS) consists of the brain in the head capsule and the ventral nerve cord (VNC; also known as ventral nervous system) in the thorax (*Court et al., 2017*; *Ito et al., 2014*). The VNC receives and integrates sensory input from the periphery and sends this information to the brain in ascending neurons through the cervical connective (*Tsubouchi et al., 2017*). The brain, in turn, sends sensory-motor signals to the VNC via descending neurons (*Namiki et al., 2018*). The VNC transforms these signals into locomotor actions (*Harris et al., 2015*). It controls muscles in the thorax in unique ways, depending on whether it is steering the wings during flight or generating acoustic communication signals during both reproductive and agonistic behaviors (*Clyne and Miesenböck, 2008*; *Jonsson et al., 2011*; *Shirangi et al., 2013*; *von Philipsborn et al., 2011*). It coordinates muscles in the legs to walk, jump, groom, reach, touch, and taste (*Bidaye et al., 2014*; *Card and Dickinson, 2008*; *Chen et al., 2018*; *Gowda et al., 2018*; *Harris et al., 2015*; *Howard et al., 2019*; *Kim et al., 2017*; *Mamiya et al., 2018*; *Mendes et al., 2013*; *Mendes et al., 2014*; *Seeds et al., 2014*; *Tuthill and Wilson, 2016*; *Wosnitza et al., 2013*). The VNC also controls musculature in the abdomen relevant to copulation and reproduction including abdominal bending, attachment, and ejaculation in the male (*Crickmore and Vosshall, 2013*; *Jois et al., 2018*; *Pan et al., 2011*; *Pavlou et al., 2016*; *Tayler et al., 2012*), and sperm storage and oviposition in the female (*Kimura et al., 2015*; *Lee et al., 2015*).

These different functions are orchestrated by anatomically discrete segments of the VNC. Whereas the thoracic neuromeres control the legs and wings, the abdominal neuromere controls the abdominal muscles, gastric system, and reproductive organs. Much of the developmental mechanisms that generate and assemble the VNC have been characterized (*Venkatasubramanian and Mann, 2019*). Like other holometabolous insects, *Drosophila* undergo two stages of neurogenesis building two distinct nervous systems. Embryonic neurogenesis gives rise to the larval nervous system, and post-embryonic neurogenesis produces adult specific neurons. The structure of the adult nervous system is established in the embryo where pioneering neurons setup networks of tracts,

which later developing neurons follow (*Hartenstein, 2018*). The adult VNC is comprised of 4 primary neuromeres, three thoracic (one per each pair of legs: the prothoracic neuromere, ProNm; mesothoracic neuromere, MesoNm and metathoracic neuromere, MetaNm) and one fused abdominal neuromere (ANm) (*Court et al., 2017*; *Niven et al., 2008*). The neuromeres, composed of one or more fused neuropils, are segment-specific parts of the CNS which process sensory signals and control movements of their specific segments. The thoracic neuromeres are homologous structures and are thus morphologically similar to each other (*Smarandache-Wellmann, 2016*). The neurons in these segments are derived from a set of repeating type I neuroblasts. Cells derived from a given neuroblast produce a lineage, which can be split into Notch$^+$ and Notch$^-$ hemilineages (*Truman et al., 2010*). The formation of tagmatic boundaries, which group these segments into morphological units along the body axis, is established through differential expression of Hox family transcription factors (TFs) (*Angelini and Kaufman, 2005*). Cell types within each neuromere are genetically encoded by developmental programs. However, it is unclear whether the mature terminal identity of an individual neuron can be determined from its adult transcriptome.

Although a large body of work has investigated how the adult *Drosophila* CNS is established from that of the embryo, many outstanding questions remain. It is for example unknown whether the TFs involved in establishing the cellular diversity of the nervous system also play a role in the form and function of these same neurons in the adult. Although certain features of individual neurons can be plastic, the identity of a terminally differentiated neuron is likely to remain stable throughout the animal's life. It is therefore important to understand gene expression and regulation that permit mature neurons to preserve their subtype identity, morphology and connectivity, and maintain neural circuit function throughout adulthood. The VNC of the genetically tractable vinegar fly, *Drosophila melanogaster*, is ideally suited for these studies.

Here we used single-cell RNA-sequencing to characterize the transcriptomes of individual cells in a 5 day old adult *Drosophila* VNC. We generated an atlas of the VNC with 26,768 single-cell transcriptional profiles that define more than 100 cell types. This analysis reveals that the VNC has a roughly equal number of inhibitory GABAergic neurons and excitatory cholinergic neurons, and that genes encoding prepropneuropeptides are amongst the most highly expressed. The segmentally repeating nature and developmental history of the VNC is born out in the cellular transcriptomes, as maintained expression of Hox and several other neuronal lineage marker genes persist. In addition, these profiles provide many novel markers for known and new cell types. This single-cell atlas of the adult *Drosophila* VNC provides a useful resource to help connect cellular identity to behaviorally relevant neural circuit function.

## Results

### Single-cell transcriptomic atlas of the adult VNC

We created a single-cell transcriptome atlas of the VNC using single-cell RNA-sequencing with 10x Chromium chemistry. We processed 80 VNCs, 20 per independent replicate (*Figure 1A*; see Materials and methods). The data were filtered according to the number of genes (nGene), the number of unique molecule identifiers (nUMI), and the proportion of mitochondrial genes (prop.mito) (*Figure 1—figure supplement 1A,B*; see Materials and methods). A similar number of expressed genes, and UMIs were retrieved in each replicate (*Figure 1—figure supplement 1C*). Pooling the data, we recovered 26,768 high quality cells with a median of 1170 genes and 2497 transcripts per cell. We compared the expression levels observed in this single-cell data set with those reported from bulk sequencing, to evaluate the extent to which the tissue dissociation and the 10x procedure effect gene expression. The filtered single-cell data showed high correlation (r = 0.76) to FlyAtlas' bulk VNC sequencing data (*Figure 1—figure supplement 1D*; *Leader et al., 2018*). Comparisons of the pseudo-bulk expression levels between replicates of the filtered cells yielded correlations coefficients from 0.93 to 0.95 (*Figure 1—figure supplement 2*).

We performed a Canonical-Correlation Analysis (CCA) on the transcriptomes and reduced the first 45 dimensions into two t-SNE dimensions (*van der Maaten, 2014*; *Figure 1B*). We evaluated the clustering resolution with a clustering tree, comparing the relationship between clusters at different resolutions (*Figure 1—figure supplement 3*). A cluster resolution of 12 was chosen as it best resolved established substructures within our data (as described below). This resulted in 120 distinct

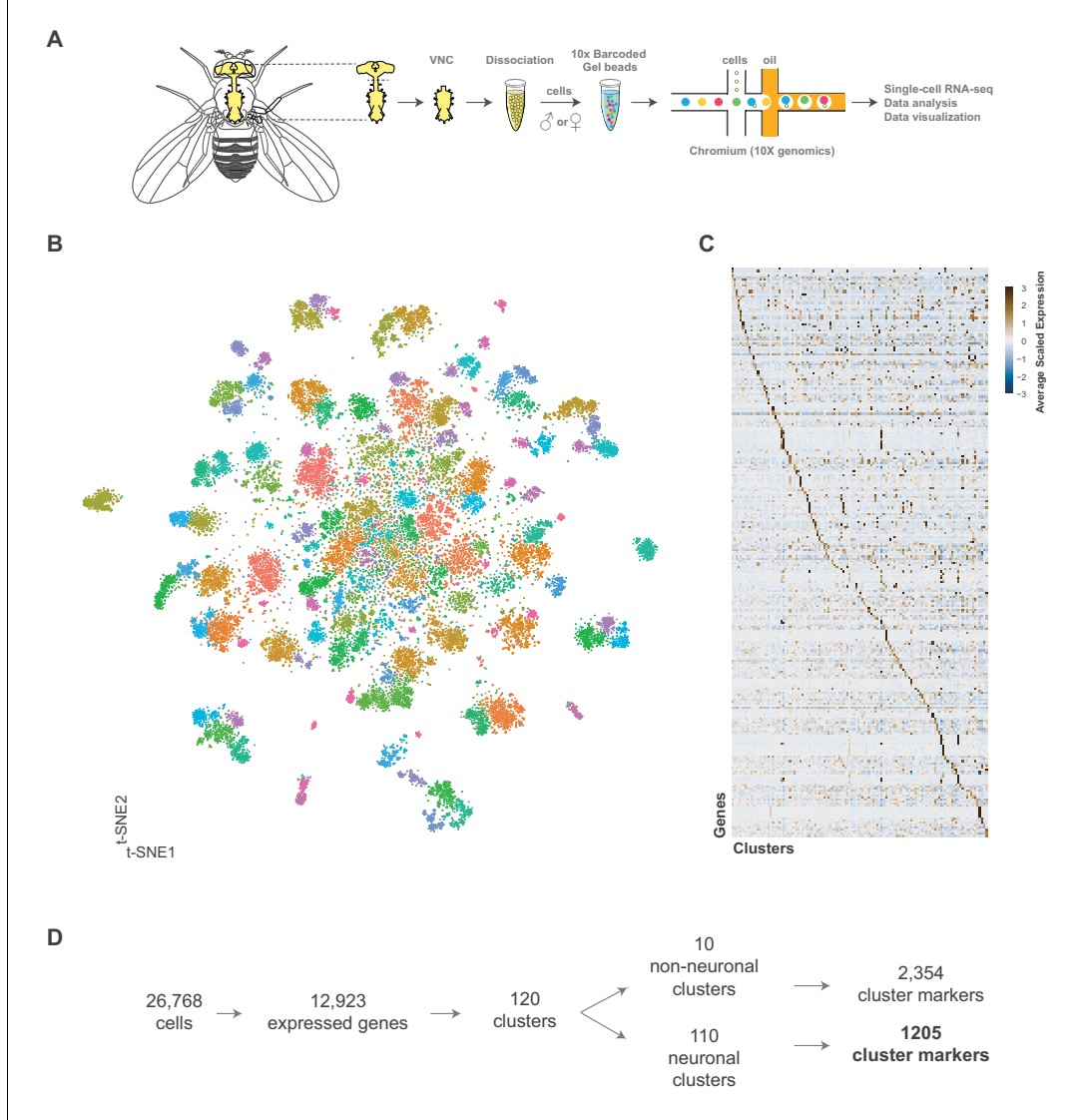

**Figure 1.** Single cell sequencing of 5 day old adult *Drosophila* VNC. (**A**) Schematic workflow showing single cell sequencing data generation. Male and female *Drosophila* VNCs were dissected and dissociated prior to droplet encapsulation of individual cells with barcoded beads, forming gel beads in emulsion (GEMs). Following barcode incorporation, molecular amplification and sequencing the transcriptional profiles of individual cells were determined. (**B**) Two-dimensional representation (t-SNE) of 26,768 *Drosophila* VNC cells grouped into 120 clusters. Clusters were assigned using the Louvain algorithm, using 45 CCA dimensions with a cluster resolution of 12. Each dot is a cell colored by cluster identity. (**C**) Heatmap showing scaled, log-normalized expression of top 5 cluster discriminative genes per cluster. (**D**) Flow diagram representing neuronal and non-neuronal cluster identification, including total number of genes identified as cluster markers.

The online version of this article includes the following source data and figure supplement(s) for figure 1:

**Source data 1.** List of marker genes for the 120 clusters shown in *Figure 1B* and *Figure 1—figure supplement 4*.
**Source data 2.** Cluster summary table.
**Figure supplement 1.** Cut-offs and correlation to bulk sequencing.
**Figure supplement 2.** Pseudo-bulk comparisons between replicates of filtered VNC data.
**Figure supplement 3.** Cluster resolution analysis of adult VNC dataset.
**Figure supplement 4.** t-SNE plot of 5 day old adult VNC with cluster numbers labeled.
**Figure supplement 5.** Replicate contributions.
**Figure supplement 6.** t-SNE spatial distributions.
**Figure supplement 7.** Identification of neuronal and glia clusters.

clusters (*Figure 1—figure supplement 4*). Cells from independent experimental replicates were proportionally distributed across the clusters with an alignment metric of 94.6% (*Figure 1—figure supplement 5A,B*). All replicates contributed to all clusters (with one exception) and showed high correlation of cluster level expression (*Figure 1—figure supplement 1C*). Replicate one did not have any constituents in cluster 118, which had only 22 cells total. The number of genes and transcripts showed a uniform distribution across the t-SNE (*Figure 1—figure supplement 6A*).

These 26,768 cells represent an approximate 1.5x coverage of the adult VNC. Lineage based counts of the adult specific post-embryonic neurons in the VNC range from 10,000 to 11,000 cells (*Birkholz et al., 2015*; *Lacin et al., 2019*), with a total number of cells in the VNC forecasted to be ~16,000 (*Lacin et al., 2019*). Connectomic analyses of the adult VNC using electron microscopy estimated a larger total number of ~20,000 cells (*Bates et al., 2019*). Based on these prior numbers, our cell atlas has 1.3–1.7x coverage of the VNC.

## Cluster-defining marker genes

The 120 t-SNE clusters are defined by unique combinations of significantly enriched genes that we refer to as cluster markers (*Figure 1C*, *Figure 1—source data 1*). The number of significantly enriched genes and the maximum observed log fold-change varies by cluster (*Figure 1—figure supplement 6B*; *Figure 1—source data 2*). These cluster markers are likely to be important for the development and/or maintenance of the cell types represented by the clusters. We compared the significantly enriched cluster markers from this data set (excluding the salivary and sperm clusters, see below) to the cluster markers of re-analyzed mid-brain Drop-seq (*Croset et al., 2018*) and whole brain 10x (*Davie et al., 2018*) data sets (*Figure 2—figure supplement 1*; *Figure 2—figure supplement 1—source data 1*; *Figure 2—figure supplement 1—source data 2*). 85% of the mid-brain cluster markers were also found in our VNC cluster markers, while 67% of the whole brain markers overlapped. This suggests that the majority of the genes that define neuronal and glial identity in the brain also define neuronal and glial identity in the VNC.

We defined 110 neuronal clusters using expression of known neuron-specific and neuron-enriched genes (*Figure 1—figure supplement 7*; see methods), such as *embryonic lethal abnormal vision* (*elav*), *neuronal Synaptobrevin* (*nSyb*) and the long non-coding (lnc) RNA *noe* (*noe*) (*Davie et al., 2018*; *DiAntonio et al., 1993*; *Kim et al., 1998*; *Robinow et al., 1988*). These neuronal clusters can be distinguished from each other by differential expression of 1205 additional cluster markers (*Figure 1D*, *Figure 1—source data 1*). Functional enrichment analysis of these cluster-defining marker genes identified those encoding immunoglobulin (Ig)-like domains as most significantly enriched (*Figure 2A*), including many cell adhesion molecules that are known to establish neuronal connectivity during development (*Özkan et al., 2013*). Specifically, many members of the Immunoglobulin superfamily (IgSF) continue to be expressed in the adult and contribute to neuronal cluster identity (*Figure 2B*; *Figure 2—figure supplement 2A*). The defective proboscis extension response (Dpr) subfamily, and their binding partners, the Dpr-interacting protein (DIP) subfamily, are markers for over 50% of neuronal clusters identified in the adult VNC (*Figure 2—figure supplement 2B*). Dpr and DIP genes provide a complex interaction network regulating neural circuit assembly and have been proposed to act as neuronal 'identification tags' during development. Distinct combinations of Dpr and DIP proteins are expressed by different neuronal classes in the developing optic lobe, the antennal lobe, and the VNC (*Carrillo et al., 2015*; *Özkan et al., 2013*). Specific combinations of Beat and Side IgSF proteins play an important role in the sculpting the nervous system (*Özkan et al., 2013*; *Pipes et al., 2001*). Almost all Beat and Side family members were found as cluster markers in the VNC (*Figure 2B*), and their enrichment was largely mutually exclusive, with 80% of the identified clusters enriched for either Beats or Sides (*Figure 2—figure supplement 2C*). The enrichment of IgSFs in the adult VNC supports their importance in establishing cell type specificity and suggests an ongoing role in maintenance of neuronal connectivity in the mature nervous system (*Figure 2B*).

G-protein coupled receptors (GPCRs) were also significantly over-represented as cluster markers (*Figure 2A*, *Figure 2—figure supplement 3A*). GPCRs play a critical role in intercellular communication by interacting with a diverse group of extracellular ligands such as neurotransmitters, modulatory neuropeptides and biogenic amines, peptide hormones, and gases (*Hanlon and Andrew, 2015*). We observed a median of 8 GPCRs expressed per cell and a median of 22 transmembrane receptors in total per cell. Due to the potential for drop-out events (a gene is expressed in a cell,

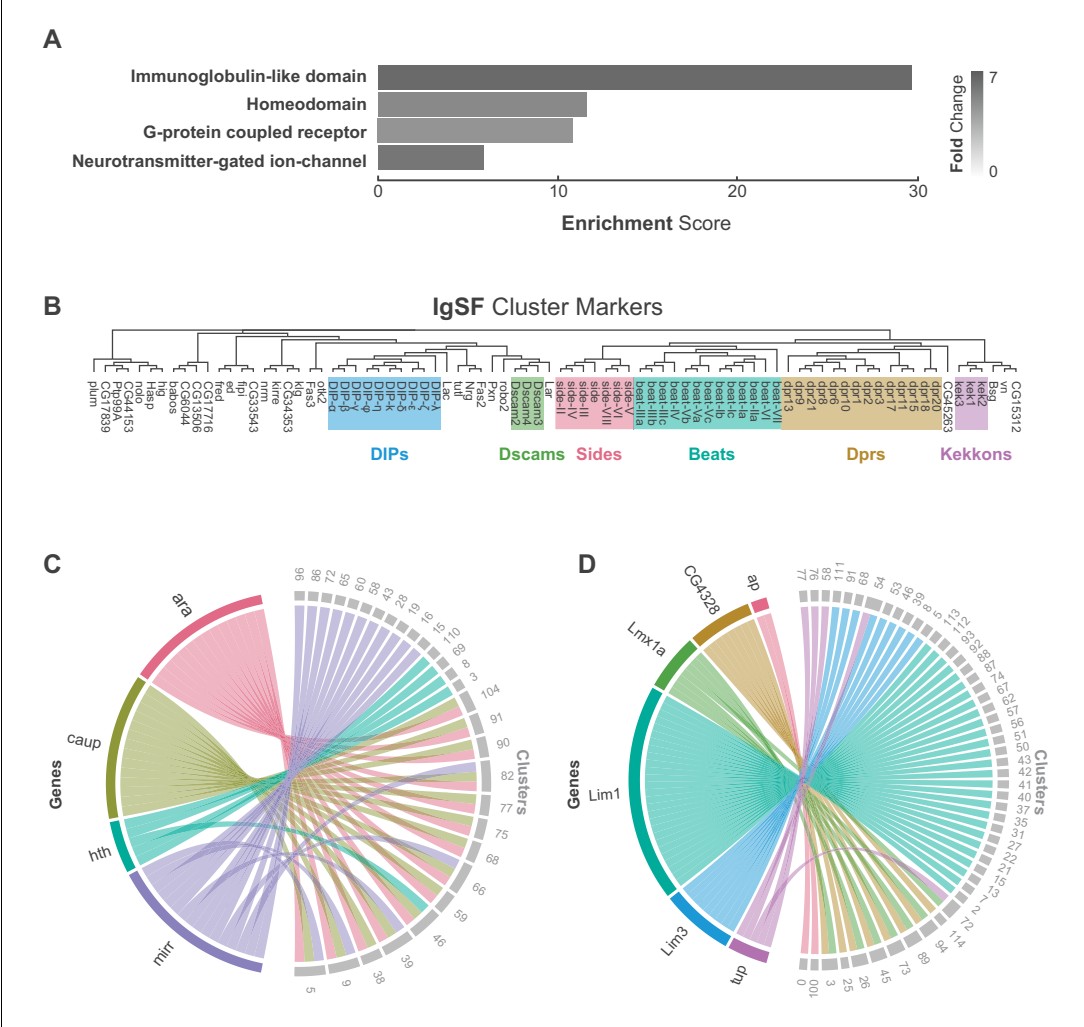

**Figure 2.** Characterization of neuronal cluster markers. (**A**) Functional analysis of neuronal cluster markers (DAVID) showing representative functional terms (>5 fold-change) and their corresponding enrichment scores (representing mean p-values). (**B**) Phylogenetic tree of IgSF neuronal cluster markers. IgSF subfamily members are highlighted in color. (**C**) Chord diagram comparing the relationship between Tale Homeobox TF cluster markers (left) and the clusters in which they are significantly enriched (right). (**D**) Chord diagram comparing the relationship between LIM Homeobox TF cluster markers (left) and the clusters in which they are significantly enriched (right). (**B–D**) Gene were classified based on Flybase Gene Group annotations (www. flybase.org). (**C,D**) Chord diagrams are visual aids based on data available in *Figure 1—source data 1*.

The online version of this article includes the following source data and figure supplement(s) for figure 2:

**Figure supplement 1.** Comparison of central nervous system single-cell data sets.

**Figure supplement 1—source data 1.** List of marker genes for the 32 clusters shown in *Figure 2—figure supplement 1A* of re-analyzed data from *Croset et al. (2018)*.

**Figure supplement 1—source data 2.** List of marker genes for the 88 clusters shown in *Figure 2—figure supplement 1B* of re-analyzed data from *Davie et al. (2018)*.

**Figure supplement 2.** Relationships between IgSF subfamilies in defining cluster identity.

**Figure supplement 3.** GPCR neuronal cluster markers.

**Figure supplement 4.** Co-expression of gene family members in the VNC.

**Figure supplement 5.** Ion channel neuronal cluster markers.

**Figure supplement 6.** TF neuronal cluster markers.

but not detected) these values may be an underestimate. We saw a median of 12 GPCRs when we consider the expression at the cluster level (with a threshold of 20% positive cells per cluster; *Figure 2—figure supplement 4B*). Although many GPCRs are significantly enriched in the individual clusters, they were, in general, expressed at low levels, in few cells per cluster, and across many

clusters (*Figure 2—figure supplement 3A*). The neuropeptide receptor family members had more inter-cluster variation in expression levels than the other GPCR subtypes, suggesting that they confer an added level of specialization. We also saw that 80% of biogenic amine receptor family members, a diverse group of neurotransmitter receptors sensitive to biogenic amine neurotransmitters including dopamine, octopamine, serotonin and tyramine, are found as cluster markers (*Figure 2—figure supplement 3B*). Odorant receptors, gustatory receptors, ionotropic receptors, and pickpocket sodium channels are predominantly found in first-order sensory neurons (*Joseph and Carlson, 2015*) and in general, were not expressed in the VNC, as expected. There were exceptions, however, as *Or63a, Gr28b, Ir76a, ppk31* are all expressed in the VNC, albeit at low levels and in few cells. Neurotransmitter-gated ion-channels were also highly enriched as neuronal cluster markers (*Figure 2A*, *Figure 2—figure supplement 5*, *Figure 1—source data 1*), including 8 of the 10 nicotinic acetylcholine receptor subunits (*Littleton and Ganetzky, 2000*). Expression of transmembrane receptors that provide fast- and slow-acting responses to neurotransmitters, therefore, contribute to cellular identity.

Finally, over 20% of all *Drosophila* TFs were cluster markers in the adult VNC, the most enriched class of which was the homeodomain (HD) family (*Figure 2A*, *Figure 2—figure supplement 6*). Homeodomain TFs play central roles in establishing regional-, tissue- and cell-specific fates (*Bürglin and Affolter, 2016*). We observed a median of 6 HD TFs per cell, and a median of 52 TFs in total per cell. We saw a median of 10 HD TFs when we consider the expression at the cluster level (20% positive cells per cluster threshold; *Figure 2—figure supplement 4D*). We detected high variance in HD TF expression levels across clusters (*Figure 2—figure supplement 6A*). Four members of the three amino acid loop extension (TALE) class of HD TFs mark specific clusters. This included all the three members of the evolutionarily conserved *Iroquois* gene complex (Iro-C) *araucan* (*ara*), *caupolican* (*caup*), and *mirror* (*mirr*) (*Cavodeassi et al., 2001*). The Iro-C arose through two duplication events, one ancient event in arthropods led to independent *ara/caup* and *mirr* genes, followed by a more recent event in dipterans which gave rise to the *caup* and *ara* genes (*Kerner et al., 2009*). Cluster expression of Iro-C genes appears to reflect this evolutionary history. Whereas *caup* and *ara* overlap as cluster markers, *mirr* expression is only partially overlapping and *mirr* is an independent marker for 11 additional clusters, suggesting specialization (*Figure 2C*). In addition, we found largely mutually exclusive enrichment of individual members of the LIM class of HD TFs defining cell clusters in the VNC (*Figure 2D*). LIM TFs are known to specify distinct neuronal identities in the embryo (*Thor et al., 1999*). In the VNC only *Lmx1a* and *CG4328* appear to be co-expressed which given their genomic linkage likely represents a relatively recent tandem duplication event. These findings suggest that aspects of the TF-code that establishes cellular identity during development actively maintains neuronal identity throughout the life of the animal (*Deneris and Hobert, 2014*).

## Hox genes define neuromere identity

Hox genes encode homeodomain proteins that confer positional identities along the antero-posterior axis in all bilaterian animals (*McGinnis and Krumlauf, 1992*). In both vertebrates and invertebrates, the Hox family of transcription factors are known to govern key aspects of nervous system development, notably the formation of neuromuscular networks (*Philippidou and Dasen, 2013*). The neuromeres of the VNC show distinct segment-specific properties under Hox gene control (*Baek et al., 2013*; *Suska et al., 2011*).

The Hox genes *Antennapedia* (*Antp*), *Ultrabithorax* (*Ubx*), *abdominal-A* (*abd-A*), and *Abdominal-B* (*Abd-B*), which specify the thoracic, and abdominal segments of the fly VNC, define specific clusters in the single-cell atlas (*Figure 3A*). Hox gene expression patterns were significantly anti-correlated at the individual cell level, except for *abd-A* and *Abd-B* (*Figure 3B*). Similar patterns were seen with the correlation of cluster-level expression, and the patterns were consistent between replicates. *Antp* and *Ubx* are both expressed throughout the t-SNE plot, with many clusters expressing both, albeit in mutually exclusive cells, while *abd-A* and *Abd-B* expression is more restricted and overlapping (*Figure 3C*). Immunostaining the adult VNC with antibodies against these Hox proteins showed belts of expression along the anterior-posterior axis (*Figure 3D*). These patterns of expression are different to those in the larva but are consistent with those observed in the mid-pupal stage (*Baek et al., 2013*). Antp protein is most highly expressed in the mesothoracic neuromere (MesoNm), whereas Ubx expression is highest in the metathoracic neuromere (MetaNm). Abd-A and Abd-B protein showed overlapping expression restricted to the abdominal neuromere (ANm), with

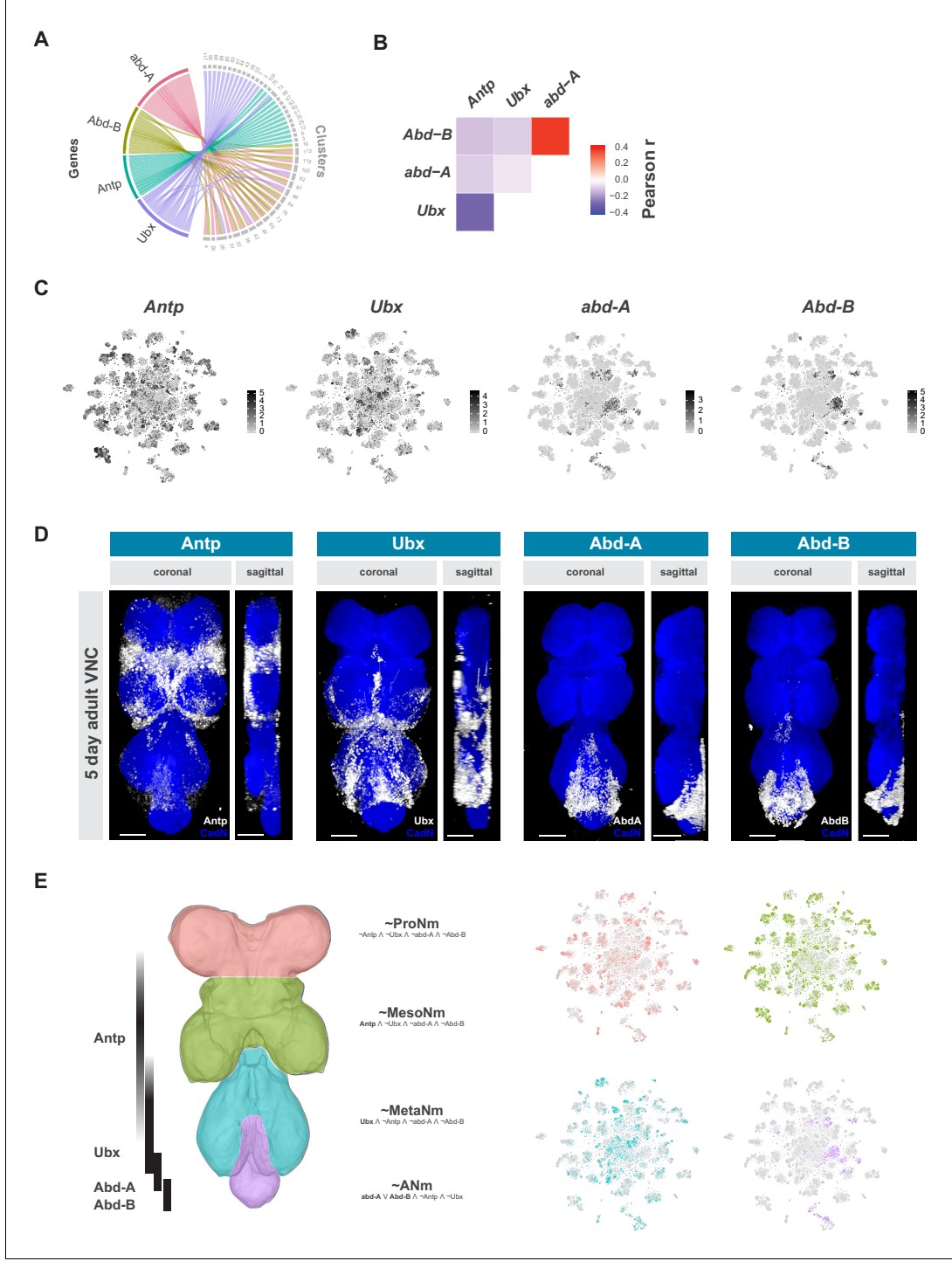

**Figure 3.** Neuromeres are defined by Hox gene expression in the adult VNC. (**A**) Chord diagram comparing the relationship between Hox genes found as cluster markers (left) and the clusters in which they are significantly enriched (right), based on data available in *Figure 1—source data 1*. (**B**) Heatmap of Pearson correlation coefficient for Hox gene expression in all single cells. We saw consistent patterns of correlation at the cluster-level expression, and between replicates. (**C**) t-SNE plot of Hox gene expression distribution. Expression shown in black, intensity is proportional to the log-normalized expression levels. (**D**) Visualization of Hox protein expression in the 5 day old adult VNC. Coronal and sagittal views of anti-Antp, -Ubx, -Abd-A, and -Abd-B (white). Neuropil counterstained with anti-Cad-N (blue). Scale bars = 50 µm. (**E**) Schematic representing bands of Hox expression along the anterior-posterior axis of the VNC (left). Cells assigned to approximate neuromeres in VNC t-SNE plots (right) based on differential Hox gene expression: ProNm (pink), MesoNm (green), MetaNm (blue) and ANm

*Figure 3 continued on next page*

*Figure 3 continued*

(purple). Expressed genes used in defining each neuromere are in bold. (symbols represent the following: '¬' not, '∧' and, '∨' or, '~' approximate).

Abd-A expressed more anteriorly than Abd-B. By combining the neuromere enriched expression revealed by antibody staining, with the high level of anti-correlation in the single-cell data, we can separate the single-cell clusters into approximations for each neuromere (*Figure 3E*).

## Neuroblast lineage identity

Each neuromere contains a repeating set of neuroblast lineages. We used prior knowledge of biomarkers defining the development of post-embryonic lineages (*Bossing et al., 1996*; *Schmid et al., 1999*) to further annotate the 5 day old adult VNC single-cell atlas. All known lineage biomarkers showed continued expression in the adult VNC, and most were found as significantly enriched cluster markers (*Figure 4A and B*; *Figure 4—figure supplement 1*; *Figure 1—source data 1*). These established markers therefore allowed us to assign potential hemilineage identities to many of the cell clusters (*Figure 4B*; *Figure 1—source data 2*). We also used fast-acting neurotransmitter (FAN) identity as an additional marker to specify potential hemilineages, since FAN usage is acquired in a lineage-dependent manner in post-embryonic neurons in the VNC (*Lacin et al., 2019*). For example, cluster 54 has enriched expression of lineage markers *Lim3*, *tailup* (*tup), abnormal chemosensory jump 6* (*acj6*), and the glutamatergic neuron marker *Vesicular glutamate transporter* (*VGlut*), which collectively suggests this cluster is hemilineage 9B. The correspondence between cluster markers and hemilineage markers shows that cell cluster identity is closely linked to hemilineage identity. We observed a strong concordance between the number of cells within our predicted hemilineages and previous cell counts (*Figure 4—figure supplement 2A*; *Birkholz et al., 2015*; *Lacin et al., 2019*). This correlation of cell counts further supports our atlas having an approximately 1.5x coverage of the VNC.

Many of the established lineage markers are expressed at low levels and in few cells (e.g., *B-H1/2*, *exex*, *tup*), while others are robustly expressed (e.g., *acj6*, *toy*, *Lim3*) (*Figure 4—figure supplement 1*). Nevertheless, the restricted cluster-specific expression of these established markers and co-expression with more robust novel markers (*Figure 4C*; *Figure 4—source data 1*) allowed us to assign cells to putative hemilineages. At present some hemilineages cannot be defined with certainty due to the lack of specific sets of established markers. For instance, there is no marker for hemilineage 2A, other than *VGlut*, and hemilineage 3B's *Dbx* expression does not persist past the pupal stage (*Lacin and Truman, 2016*; *Lacin et al., 2019*). Similarly, *unc-4* is currently the sole established marker for many cholinergic hemilineages, including 7B, 12A, 18B and 19B. We observe multiple cholinergic clusters enriched for *unc-4* (*Figure 1—source data 1*; *Figure 4—figure supplement 1*). To assign these clusters to known hemilineages, additional cluster specific enriched genes can be investigated by immunohistochemistry. Conversely, clusters 6, 69, and 110 express markers for both hemilineages 8A (*ems*, *ey*, *VGlut*) and 24B (*ems*, *toy*, *VGlut*), suggesting that these lineages may exhibit very similar transcriptional profiles and resolving them will likely require more cells (*Figure 4C*).

We have identified several novel potential hemilineage marker genes (*Figure 4—figure supplement 1*). Of the 59 most highly enriched hemilineage-defining genes, shown in *Figure 4C*, 34 are TFs, approximately half of which contain homeodomains. Interestingly, 5 of these novel marker genes encode lncRNAs. lncRNAs have been shown to have complex spatiotemporal regulation throughout the embryo (*Wilk et al., 2016*; *Karaiskos et al., 2017*) and specifically during embryonic neurogenesis (*Scruggs et al., 2015*), suggesting their functional importance during development. Two of these lncRNAs, *MRE23* and *CR45141*, are adjacent to and transcribed divergently to their nearest neighbour protein coding genes, *fork head* (*fkh*) and *vestigial (vg)*, respectively. These gene pairs show clear positive correlations of expression within the hemilineages (*Figure 4C*), suggesting they share a cis-regulatory landscape. Such divergent transcription has been shown to enhance transcriptional output (*Scruggs et al., 2015*; *Schor et al., 2018*). As the roles of many of these novel marker genes have yet to be studied either during development or in the adult nervous system, our VNC analysis provides a valuable resource for future investigation of hemilineage cell types.

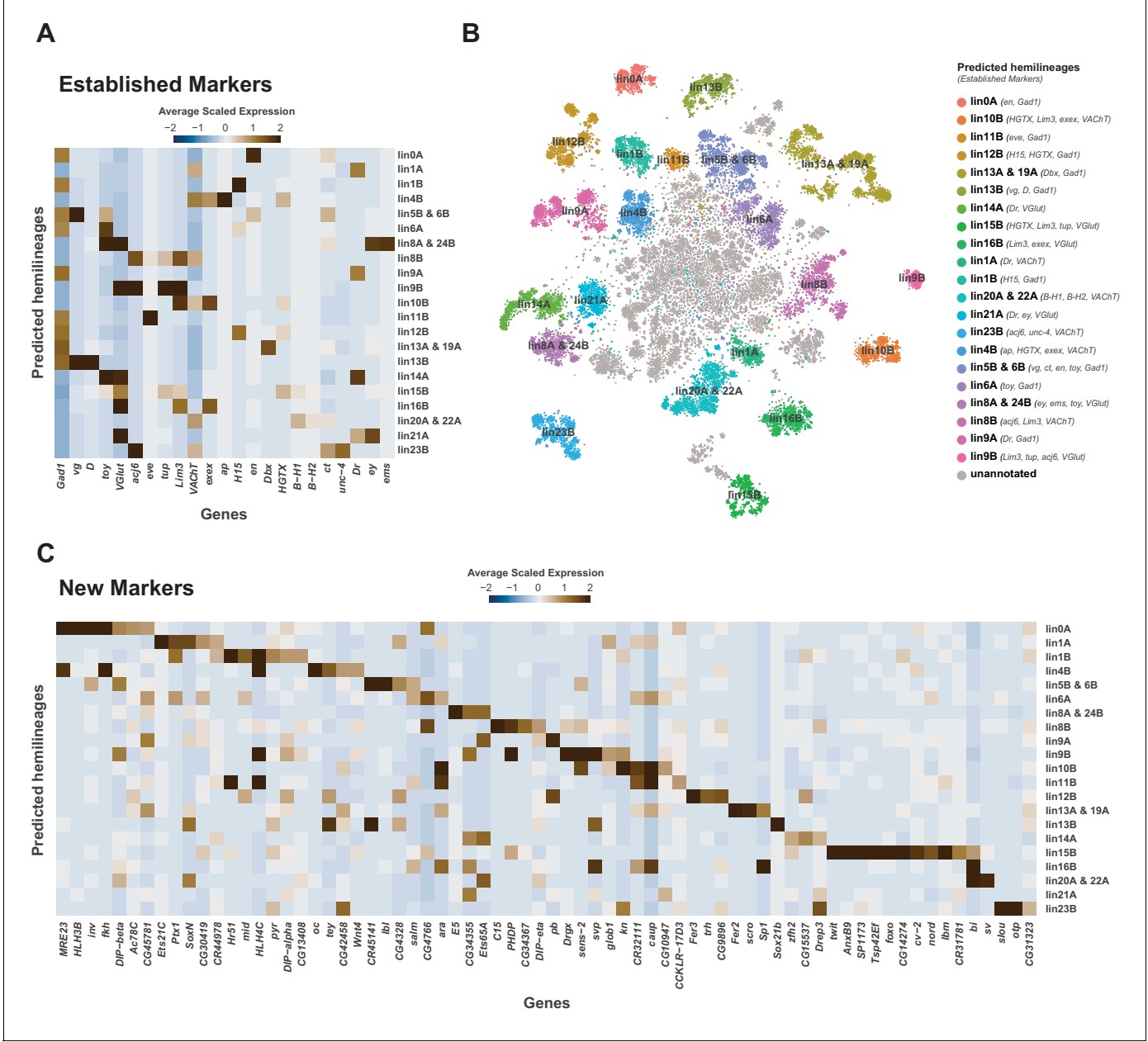

**Figure 4.** Cellular identities defined by developmental lineages. (**A**) Heatmap of the mean scaled log-normalized expression of established lineage markers genes (bottom) within predicated neuroblast hemilineage (right). (**B**) t-SNE plot of neuronal cells highlighting predicted hemilineages based on the expression of established hemilineage marker genes (shown beside list of predicted hemilineages on the right). (**C**) Heatmap of the mean scaled log-normalized expression of potential new hemilineage markers genes (bottom) within predicated neuroblast hemilineage (right). Full list of predicted hemilineage markers can be found in *Figure 4—source data 1*.

The online version of this article includes the following source data and figure supplement(s) for figure 4:

**Source data 1.** List of marker genes for predicated hemilineages shown in *Figure 4B*.

**Figure supplement 1.** Neuroblast lineage marker gene expression in the VNC.

**Figure supplement 2.** Neurodevelopment gene expression.

Many predicted hemilineages contain multiple clusters (*Figure 4B*; *Figure 1—source data 2*). We propose that these clusters represent distinct variations within a hemilineage, partly informed by birth order and partly related to thoracic vs. abdominal anatomical position. Consistent with a subdivision based on birth order, many of our predicted hemilineages contain a distinct subgroup of

broad (*br*) expressing cells (*Figure 4—figure supplement 2B*). A specific isoform of *br* continues to be expressed in the adult and marks cells born around L2-L3 ecdysis (*Zhou et al., 2009*). Other hemilineage sub-division was evident when visualizing expression of neurodevelopmental genes such as *prospero* (*pros*), *datilografo* (*dati*), *maternal gene required for meiosis* (*mamo*), and *IGF-II mRNA-binding protein* (*Imp*) (*Figure 4—figure supplement 2C*). Expression of *pros* and *Imp* in the VNC were negatively correlated (*Figure 4—figure supplement 2D*), as previously observed in the adult brain (*Davie et al., 2018*). In contrast, expression of *pros* was highly correlated with that of *dati*, while *Imp* and *mamo* were highly correlated, suggesting these pairs of genes may be co-regulated in the adult (*Figure 4—figure supplement 2D*). *pros* and *Imp* also showed an anatomical bias, with *pros* being negatively correlated with abdominally expressed *abd-A* and *Abd-B*, while *Imp* is positively correlated with these markers (*Figure 4—figure supplement 2E*). However, we did not observe a striking physiological bias, as neither *pros* nor *Imp* showed correlation with fast-acting neurotransmitter markers (*Figure 4—figure supplement 2F*). We saw consistent patterns of correlation at the cluster-level expression, and between replicates. It has previously been shown that *pros* labels late-born motor neurons from hemilineage 15B (*Baek et al., 2013*), and that a trade-off of *Imp* and *pros* expression is essential for cell cycle exit of post-embryonic type I neuroblasts in the VNC (*Maurange et al., 2008*; *Yang et al., 2017*). Studies investigating the transcriptomes of mature (embryonic) vs. immature (post-embryonic) neurons in the larval VNC have shown that *Imp* is enriched in embryonic neurons, and *pros* is enriched in post-embryonic neurons (*Etheredge, 2017*). We propose that the continued expression of *Imp* in the adult VNC labels early-born and embryonic neurons while *pros* labels late-born post-embryonic neurons.

## Gene expression defines neuronal subtypes within a hemilineage

Hemilineage 23B marked by *acj6*, *unc-4*, and *VAChT* has four distinct clusters that can only be partially explained by potential birth order and neuromere identity (*Figure 3*; *Figure 4*). *acj6* is also a marker for the cholinergic hemilineage 8B and the glutamatergic hemilineage 9B (*Figure 4A*; *Lacin et al., 2019*). We examined *acj6* expressing cells in more detail by visualizing the anatomical distribution of *acj6* expressing cell bodies and neuronal projections in the VNC using *acj6*$^{GAL4}$ to express the GAL4-responsive dual reporter *UAS-Watermelon* (WM), which simultaneously marks the plasma membrane with GFP and the cell nucleus with mCherry (*Figure 5A*; *Lee et al., 2018*). We also validated the accuracy of *acj6*$^{GAL4}$ using an anti-Acj6 antibody (*Clyne et al., 1999*; *Figure 5—figure supplement 1*). Consistent with previous reports, *acj6*$^{GAL4}$ labeled distinct clusters of cells in the three thoracic neuromeres of the VNC, which predominantly innervate the leg neuropils. (*Figure 5A*; *Harris et al., 2015*; *Shepherd et al., 2019*). The *acj6* expressing neurons were confirmed as hemilineages 8B, 9B, and 23B (D Shepherd, pers. comm.) on the basis of their primary neurite projections (*Shepherd et al., 2016*).

The *acj6* expressing clusters in the single-cell data set are highlighted in *Figure 5B*. Based on co-expression of additional markers, we could assign *acj6*-expressing hemilineages within the VNC data; lin8B expresses *acj6*, *Lim3*, and *VAChT*; lin9B *acj6*, *Lim3*, *VGlut*, and *tup*; Lin23B *acj6*, *unc-4*, and *VAChT* (*Figure 5C*). Of the 4 distinct clusters within predicted hemilineage 23B, the genes *knot* (*kn/collier*) and *tiwaz* (*twz*) (*Crozatier et al., 1996*; *Williams et al., 2014*) are significantly enriched in two independent clusters; 51 and 93, respectively (*Figure 5D*; *Figure 1—source data 1*). We validated the co-expression evident in the single-cell data using GAL4 lines that represent *kn* and *twz* expression, and co-stained these VNCs with anti-Acj6 antibody (*Figure 5E and F*; *Figure 5—figure supplement 2*). We also used a reporter for the established marker *Lim3*, which is co-expressed with *acj6* in hemilineages 8B and 9B, but not 23B (*Figure 5B*; *Lacin and Truman, 2016*). Hemilineage 23B is located dorsolaterally at the posterior edge of each of the three thoracic neuromeres, whereas 8B and 9B are both located laterally at the anterior side of each thoracic neuromere with 8B located near the ventral surface and 9B located more dorsally (*Figure 5E*; *Lacin et al., 2019*; *Shepherd et al., 2019*). Focusing on the ProNm-MesoNm border, which encompasses prothoracic 23B and mesothoracic 8B and 9B, we found that a subset of 23B co-expressed *kn* and a subset co-expressed *twz* (*Figure 5F*; *Video 1*; *Video 2*). Of the 40 dorsolateral Acj6$^+$ 23B cells, 14 were *kn*$^+$ and 9 were *twz*$^+$ (*Figure 5—source data 1*; *Figure 5—figure supplement 3*). These proportions (34% and 21%) are roughly consistent with that seen in the single-cell data (31% and 18% for *kn* and *twz*, respectively). *Lim3* expression was restricted to 8B and 9B, with 90% of Acj6$^+$ 8B and 9B cells expressing *Lim3*, and 0% co-expression in hemilineage 23B (*Figure 5F*; *Figure 5—source data 1*;

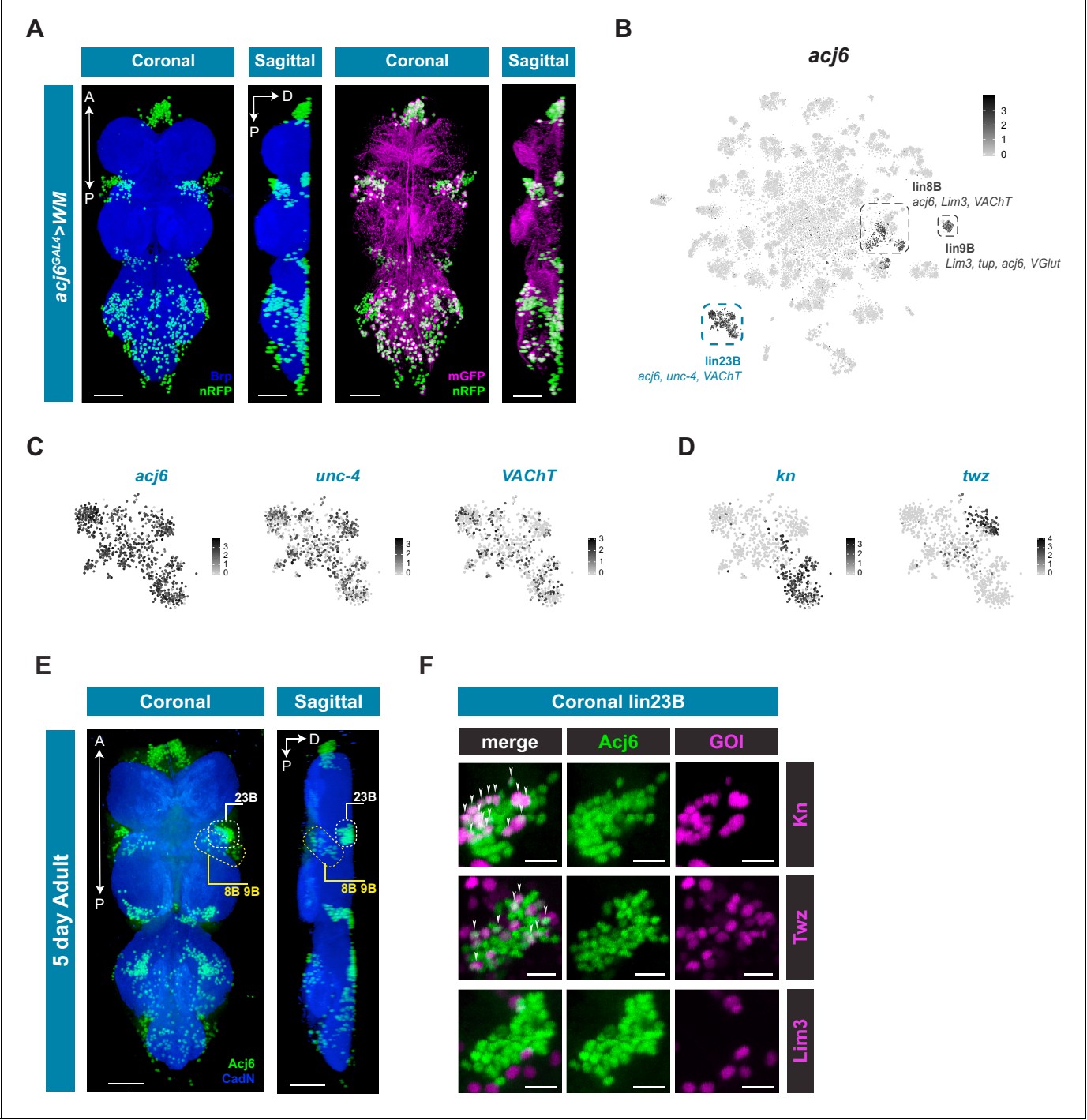

**Figure 5.** *acj6* expression, hemilineage identity, and novel sub-lineage marker co-expression in the VNC. (**A**) Visualization of *acj6* expressing cells in the 5 day old adult VNC. Maximal coronal projection, and hemi-maximal sagittal projectionof *acj6^GAL4* driving expression of *UAS-WM*, enabling the visualization of cell nuclei (nRFP; green) and neuronal morphology (mGFP; magenta). Neuropil is counterstained (Brp; blue). A, Anterior, P, posterior, D, dorsal; scale bars = 50 μm. (**B**) t-SNE plot of *acj6* expression in the VNC. Predicted hemilineage 8B, 9B, and 23B, based on known markers (shown below) are highlighted. (**C**) t-SNE plots of predicted hemilineage 23B, showing expression of markers *acj6*, *unc-4*, and *VAChT*. (**D**) t-SNE plots of predicted hemilineage 23B, showing expression of novel sub-type markers *kn* and *twz*. (**B–D**) Expression shown in black, intensity is proportional to the log-normalized expression levels. (**E**) Maximal coronal view (left) and hemi-maximal sagittal view (right) of Acj6 expression (green) in the 5 day old adult VNC. The posterior prothoracic hemilineages 23B (dashed white box) and the anterior mesothoracic 8B and 9B (dashed yellow box) are marked. A, Anterior, P, posterior, D, dorsal; Neuropil is counterstained (Cad-N; blue) scale bars = 50 μm. (**F**) Close-up maximal coronal views of dorsolateral hemilineage 23B showing the co-expression of *kn^GAL4* and *twz^GAL4* driven *UAS-Stinger* expression (nGFP; magenta) with Acj6 (green). Co-positive cells

*Figure 5 continued on next page*

Figure 5 continued

(white) are indicated with arrow heads. *Lim3^GAL4* driven expression did not colocalize with Acj6 in hemilineage 23B. GOI = gene of interest, scale bars = 10 µm.

The online version of this article includes the following source data and figure supplement(s) for figure 5:

**Source data 1.** Cell counts of novel sub-lineage marker co-expression in the VNC.
**Figure supplement 1.** *acj6^GAL4* recapitulates Acj6 protein expression in the adult VNC.
**Figure supplement 2.** Novel sub-lineage marker expression in the VNC.
**Figure supplement 3.** Single slices of novel sub-lineage marker co-expression in the VNC.

*Video 3*; *Figure 5—figure supplement 3*). *kn* was also expressed in a subset of predicted hemilineage 9B in the single cell data (*Figure 5—figure supplement 2*), and co-expression was seen between *kn^GAL4* and Acj6 in the 8B/9B cluster *in vivo* (*Figure 5—figure supplement 3*; *Figure 5—source data 1*). Full expression patterns in the VNC are shown in *Figure 5—figure supplement 2*. These findings highlight the predictive power of the single-cell transcriptome to identify new markers that refine our understanding of hemilineage subtypes which contribute to neuronal cell diversity in post-embryonic lineages.

## Classification using fast-acting neurotransmitters

A recent comprehensive map of fast-acting neurotransmitter (FAN) usage across the VNC found that neurons within a post-embryonic hemilineage use the same neurotransmitter; either acetylcholine, gamma-aminobutyric acid (GABA), or glutamate, unifying their shared developmental and functional identities (*Lacin et al., 2019*). We examined FAN usage in the VNC using expression of established biosynthetic and vesicular loading markers: *Choline acetyltransferase* (*ChAT*) and *Vesicular acetylcholine transporter* (*VAChT*) for acetylcholine, *VGlut* for glutamate, and *Glutamic acid decarboxylase 1* (*Gad1*) and *Vesicular GABA Transporter* (*VGAT*) for GABA (*Figure 6A*). Clusters showed mutually exclusive enrichment of *VAChT*, *VGlut*, and *Gad1* (*Figure 6B*). However, this exclusivity partially breaks down at the cell-by-cell level. Although expression levels of *VAChT*, *VGlut*, and *Gad1* were negatively correlated (*Figure 6—figure supplement 1A*), we also observed significant levels of co-expression, with 31% of cells expressing at least two of these markers (*Figure 6—figure supplement 1B*). A similar pattern of co-expression was observed in single-cell data from the adult brain (*Croset et al., 2018*; *Davie et al., 2018*). However, immunostaining of the intact adult VNC suggested that cytoplasmic co-expression of these markers does not occur (*Lacin et al., 2019*). Nuclear transcriptomic profiles of neuronal subtypes from both the visual system and the mushroom body found no evidence for FAN co-release (*Davis et al., 2020*; *Shih et al., 2019*). Co-expression in the VNC may therefore represent contamination due to ambient RNA present in the cell suspension (discussed below). Since the cells cluster due to hemilineage identity and hemilineages are restricted to a single FAN, we assigned FAN identity by comparing the average expression of these FAN markers at the cluster level (*Figure 6—figure supplement 1C*; *Figure 1—source data 2*). Amongst the co-expressing cells, the expression of FAN makers is negatively correlated, and cells assigned to one FAN identity tended to show higher expression of the corresponding marker (*Figure 6—figure supplement 1D*). This is, however, not always the case, presumably due to the much higher expression levels seen for *VGlut* and *Gad1* (*Figure 6A*; *Figure 6—figure supplement 1D*). With these criteria (see methods), we estimate that the VNC is 40% cholinergic, 38% GABAergic, and 18% glutamatergic. The remaining 4% of neurons did not show a strong signature for any FAN (*Figure 6C and D*). These proportions differ to those in the adult brain (*Croset et al., 2018*; *Davie et al., 2018*), with inhibitory GABAergic neurons being much more prominent in the VNC (38% vs 15% in the adult

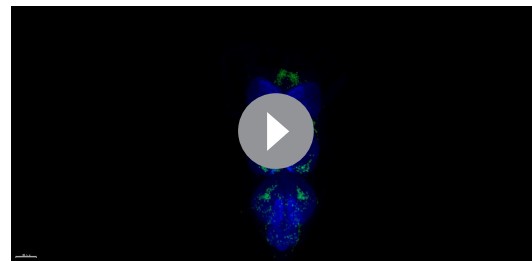

**Video 1.** Video of 3D volume showing the co-expression of Acj6 (anti-Acj6; green) with *kn^GAL4* driven expression of *UAS-Stinger* (nGFP; magenta) in the 5 day adult VNC, with neuropil counterstained (CadN; blue).
https://elifesciences.org/articles/54074#video1

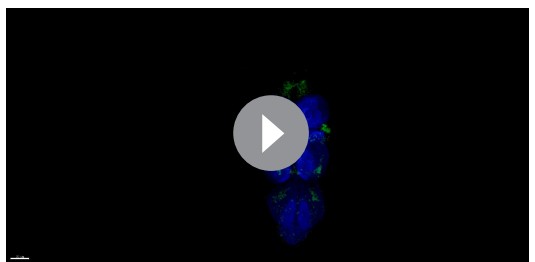

**Video 2.** Video of 3D volume showing the co-expression of Acj6 (anti-Acj6; green) with *twz*^GAL4^ driven expression of *UAS-Stinger* (nGFP; magenta) in the 5 day adult VNC, with neuropil counterstained (CadN; blue).
https://elifesciences.org/articles/54074#video2

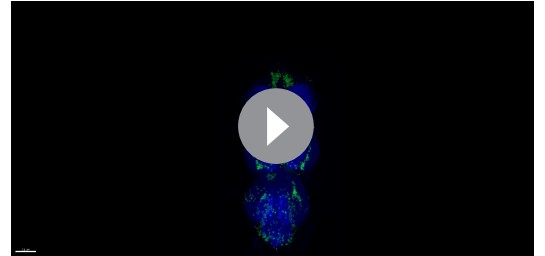

**Video 3.** Video of 3D volume showing the co-expression of Acj6 (anti-Acj6; green) with *Lim3*^GAL4^ driven expression of *UAS-Stinger* (nGFP; magenta) in the 5 day adult VNC, with neuropil counterstained (CadN; blue).
https://elifesciences.org/articles/54074#video3

midbrain). The inhibitory glutamate receptor, *GluClα* (*Cully et al., 1996*), was broadly expressed in

**Figure 6.** Fast-acting neurotransmitter usage in the VNC. (**A**) t-SNE plots showing distribution of cells expressing biomarkers for the fact-acting neurotransmitters acetylcholine (*VAChT, ChAT*), glutamate (*VGlut*) and GABA (*VGAT, Gad1*). Expression shown in black, intensity is proportional to the log-normalized expression levels. (**B**) Chord diagram comparing the relationship between fast-acting neurotransmitter cluster markers (left, genes) and the clusters in which they are significantly enriched (right, clusters). (**C**) t-SNE plot colored according to fast-acting neurotransmitter usage based on assigned cluster identity. (**D**) Percentage of cells in the VNC assigned as releasing distinct fast-acting neurotransmitters.

The online version of this article includes the following figure supplement(s) for figure 6:

**Figure supplement 1.** Expression of fast-acting neurotransmitter biomarkers in the VNC.

these data and is particularly enriched in cluster 103 (see *Figure 1—source data 1*), suggesting that a portion of glutamatergic signaling is also inhibitory. We expect this relative abundance of inhibitory signaling reflects the importance of inhibition in the initiation, maintenance, and termination of complex motor programs (*Burrows, 1992*; *Grillner, 2006*), especially in coordination between bilateral and intersegmental circuits (*Gowda et al., 2018*).

## Monoaminergic neurons

Cells that produce and likely release monoamines in the VNC can be identified based on expression of the *Vesicular monoamine transporter* (*Vmat*) gene (*Greer et al., 2005*). To examine the anatomical distribution of *Vmat* expressing nuclei and projections in the VNC we combined *Vmat$^{GAL4}$* with *UAS-Watermelon* (WM) (*Figure 7A*). *Vmat$^{GAL4}$* labels 115 ± 6.3 neurons distributed across every neuromere of the VNC (n = 9, data not shown). Each neuromere has a main cluster of cells located on the ventral surface, near the midline, consistent with known developmental origins of monoaminergic neurons. Serotonin- and dopamine-producing neurons are born from the ventromedially located paired neuroblast 7–3, and octopamine-producing neurons are born from the ventral midline unpaired median neuroblast (*Bossing et al., 1996*; *Schmid et al., 1999*). Projections of *Vmat*-labeled neurons formed thick fascicles of fibers projecting from each cluster to the dorsal surface of the VNC (*Figure 7A*, sagittal view). Each thoracic neuromere has one main fiber tract, whereas the ANm has 8 parallel tracts, one for each abdominal segment. Despite the relatively small number of *Vmat* expressing cells, they project throughout the VNC and densely innervate the entire neuropil. Additional *Vmat* nuclei associated with the VNC are consistent with its reported expression in perineural surface glia (*DeSalvo et al., 2014*). Surface glial *Vmat* expressing cells can be seen in cluster 98 of our primary t-SNE plot (*Figure 1—figure supplement 4*; *Figure 1—source data 1*).

Neurons that express *Vmat* co-cluster in the single-cell data (Clusters 72 and 84, *Figure 1—figure supplement 4*; *Figure 7B*). To define the specific monoaminergic identity of *Vmat* expressing neurons, we sub-clustered these *Vmat$^+$* cells from clusters 72 and 84 (*Figure 7C*) and identified distinct groups synthesizing specific monoamines, as defined by expression of known biosynthetic and transporter markers (*Figure 7D*; *Martin and Krantz, 2014*). All replicates contributed to all clusters and showed high correlation of cluster level expression (*Figure 7—figure supplement 1*).

## Histaminergic neurons

Histamine (HA) is well established as the primary fast neurotransmitter of adult photoreceptors in many insects, including *Drosophila* (*Hardie, 1987*; *Nässel et al., 1988*). The VNC contains 18 ventral HA-immunoreactive neurons with extensive axons and projections, six in the thoracic neuromeres and twelve in the abdominal neuromere (*Buchner et al., 1993*; *Nässel et al., 1990*). Potential new markers for HA neurons, distinguishing them from other monoaminergic neurons, include *Frequenin 1* (*Frq1*), a Ca$^{2+}$-binding protein that regulates neurotransmitter release (*Dason et al., 2009*), and *CG43795*; an uncharacterized GPCR predicted to have Glutamate/GABA receptor activity (*Agrawal et al., 2013*).

## Tyraminergic and octopaminergic neurons

Tyramine (TA) and Octopamine (OA) neurons innervate and modulate many tissues throughout the fly, including female and male reproductive systems, skeletal muscles, and sensory organs (*Pauls et al., 2018*; *Rezával et al., 2014*). *Tdc2* encoded tyrosine decarboxylase catalyzes the synthesis of TA (*Cole et al., 2005*), which can also be converted to OA by *Tbh* encoded Tyramine beta-hydroxylase (*Monastirioti et al., 1996*). All *Tdc2* expressing neurons in the VNC also express *Tbh* suggesting that they are likely to be octopaminergic, consistent with a previous report using *Tdc2$^{GAL4}$* in the VNC (*Pauls et al., 2018*). Prior work in the larval VNC suggested that the TA:OA ratio might be altered in a state-dependent manner (*Schützler et al., 2019*).

## Dopaminergic neurons

Dopamine (DA) is a critical neuromodulator controlling learning and state-dependent plasticity in the fly brain (*Cognigni et al., 2018*). Dopamine and DA neurons in the VNC have been implicated in motor behaviors such as grooming and copulation (*Crickmore and Vosshall, 2013*; *Yellman et al., 1997*). Novel DA markers in the VNC include *beat-Ib* and *beat-Ic*, a likely recent duplication in the

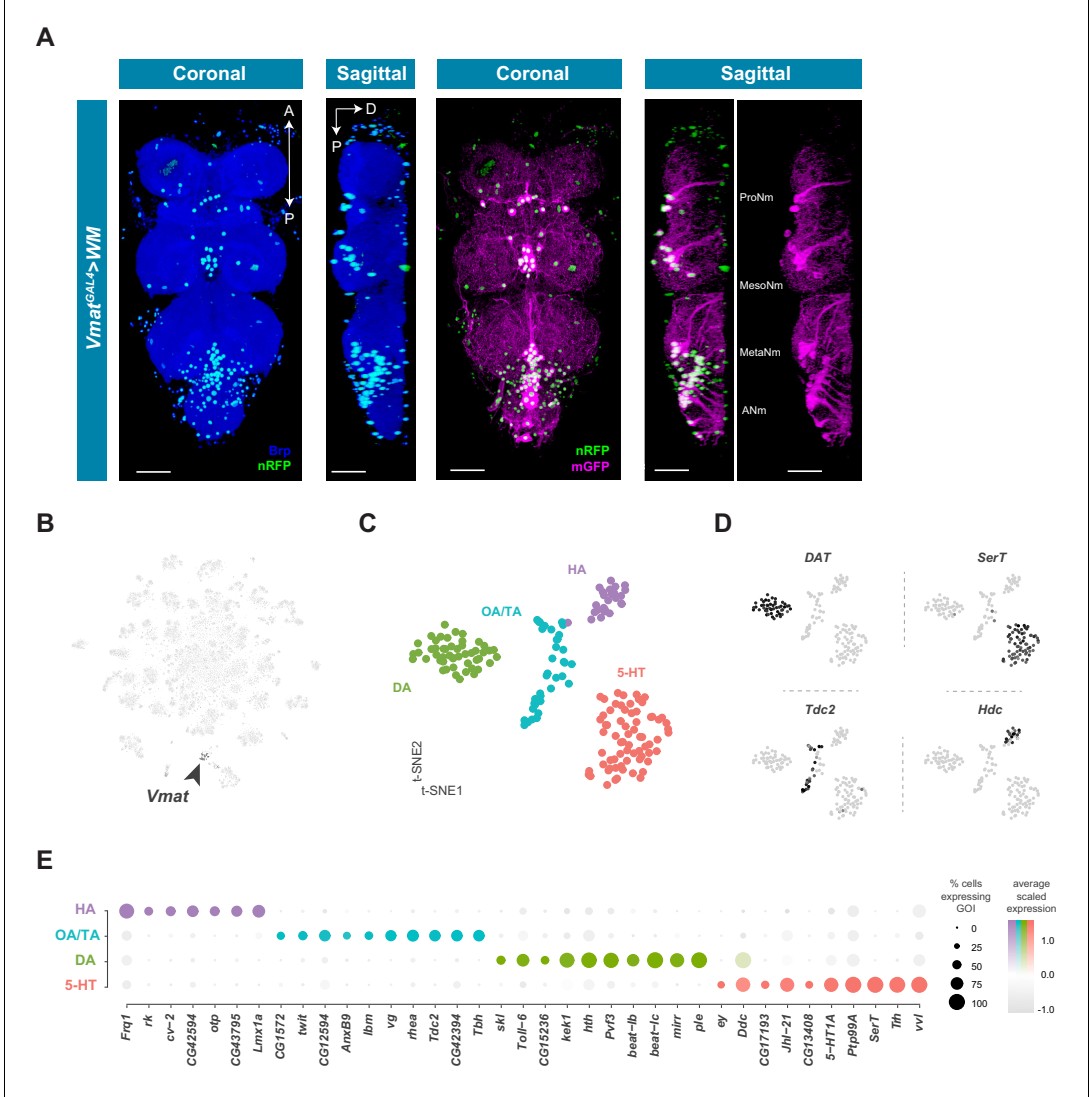

**Figure 7.** Identification and characterization of monoaminergic cell sub-types. (**A**) Visualization of *Vmat* expressing cells in the 5 day old adult VNC. Coronal and sagittal views of *Vmat^GAL4^* driving expression of *UAS-WM*, enabling the visualization of cell nuclei (nRFP; magenta) and neuronal morphology (mGFP; green). Neuropil is counterstained (Brp; blue). A, Anterior, P, posterior, D, dorsal; scale bars = 50 μm. (**B**) t-SNE plot of *Vmat* expression in black, intensity is proportional to the scaled log-normalized expression level. *Vmat* enriched Cluster 72 is highlighted with arrowhead. (**C**) t-SNE plot showing sub-clustering analysis of *Vmat*⁺ cells from clusters 72 and 84. Four sub-clusters are identified representing dopaminergic (DA), octopaminergic/tyraminergic (OA/TA), histaminergic (HA), and serotonergic neurons (5-HT). (**D**) Expression of established monoaminergic subtype-specific biomarkers used to determine cluster identity. *Histidine decarboxylase* (*Hdc*) labels Histamine (HA) neurons, *Tyrosine decarboxylase 2* (*Tdc-2*) labels Tyramine (TA) and Octopamine (OA) neurons, *Dopamine transporter* (*DAT*) labels Dopamine (DA) neurons and *Serotonin transporter* (*SerT*) labels Serotonin (5-HT) neurons. (**E**) Dot plot of the top genes in each monoaminergic sub-type based on fold-enrichment (*Figure 8—source data 1*). Size of dots represent percent of cells in cluster expressing gene of interest (GOI); intensity of color reflects average scaled expression. *Figure 7—source data 1* contain the full list of significantly enriched genes.

The online version of this article includes the following source data and figure supplement(s) for figure 7:

**Source data 1.** List of marker genes for the *Vmat*⁺ sub-clusters shown in *Figure 7C*.

**Figure supplement 1.** Individual replicate contribution to *Vmat*⁺ sub-clusters.

*beat* family of genes which act as heterophilic cell-cell adhesion molecules (*Pipes et al., 2001*). VNC DA neurons overexpress *PDGF- and VEGF-related growth factor* (*Pvf3*) and *kekkon 1* (*kek1*) which encodes a transmembrane protein that binds the EGF receptor and controls the activity of this pathway. Adult midbrain DA neurons also overexpress these genes (*Croset et al., 2018*), suggesting a possible common role for *Pvf3* and *kek1* in DA neuron function. The *Toll-6* neurotrophin-like

receptor implicated in neuronal survival and motor-axon targeting (*McIlroy et al., 2013*), appears specifically enriched in VNC DA neurons.

## Serotonergic neurons

Serotonin (5-hydroxytryptamine, 5-HT) releasing neurons within the thoracic neuromeres of the VNC modulate walking speed in a context-independent manner as well as in response to startling stimuli (*Howard et al., 2019*). In the abdominal ganglion, two clusters of sexually dimorphic neurons expressing 5-HT and *fruitless* (approximately ten cells per cluster in males) innervate the internal reproductive organs. These abdominal clusters control transfer of sperm and seminal fluid during copulation (*Billeter et al., 2006*; *Lee and Hall, 2001*; *Lee et al., 2001*). 5-HT neurons in the VNC express the 5-HT receptor *5-HT1A*, suggesting auto-regulatory/autocrine control of 5-HT neurons. VNC 5-HT neurons also express the amino acid transporter encoded by *juvenile hormone inducible 21 (JhI-21)* which is required for 5-HT-dependent evaluation of dietary protein (*Ro et al., 2016*). Finding *JhI-21* expression in 5-HT neurons in the VNC, therefore suggests that some 5-HT neurons may directly sense circulating amino acids.

## A transcription factor code for monoaminergic neurons

The top group of genes defining each subclass of monoaminergic neurons types also included unique expression of Homeobox-containing TFs (*Bürglin and Affolter, 2016*; *Figure 7E*). *orthopedia* (*otp*) and the *LIM homeobox transcription factor 1 alpha* (*Lmx1a*) define the HA cluster. DA neurons are enriched for the Hox co-factor *homothorax* (*hth*) as well as the Iroquois homeobox TF *mirror* (*mirr*). Lastly, 5-HT neurons express the homeobox TFs *ventral veins lacking* (*vvl*), *eyeless* (*ey*) and *Lim3*. Although these different combinations of homeobox genes likely act in these cells to specify position-specific patterning decisions and wiring specificity during development, it is also possible that they contribute to the function of mature monoaminergic neurons.

## Peptidergic neurons

Neuropeptides are the largest family of signaling molecules in the nervous system and act as important regulators of development, physiology, and behavior (*Nässel and Zandawala, 2019*). We analyzed neuropeptide expression in the adult VNC by looking at the expression of genes encoding neuropeptide precursors, from which active neuropeptides are derived. We identified 28 neuropeptides that are expressed in at least one cell, at a level of 10 or more transcripts per cell (*Figure 8A*). Some neuropeptides are known to be co-expressed with FANs, where they serve to increase signaling flexibility within neural networks (*Croset et al., 2018*; *Nässel, 2018*; *Nusbaum et al., 2017*). We found that in the VNC some neuropeptide genes are predominantly co-expressed with particular FANs, e.g. *sNPF* and *spab* in cholinergic neurons, *MIP* and *CCHa2* in GABAergic neurons, and *Ilp7* and *Proc* in glutamatergic neurons (*Figure 8A*; *Figure 6*). A few neuropeptides were also co-expressed in the same cells, e.g. *AstC* and *CCHa2* in GABAergic cluster 94. Some of these associations are similar to what was observed previously in mid-brain single-cell data (*Croset et al., 2018*), e.g. *sNPF* and *spab* in cholinergic neurons, but others are strikingly different. *AstC* and *CCHa2* are robustly expressed in GABAergic neurons in the VNC but are notably absent from *Gad1* expressing neurons in the mid-brain.

Some neuropeptides are also expressed in 'neurosecretory' neurons that lack FANs. Peptides released from these cells into circulation provide neuroendocrine signals, whereas those released into the nervous system have neuromodulatory function. Cells in cluster 84 exhibit several characteristics of neurosecretory cells. The TF *dimmed* (*dimm*) is required for the differentiation of neurosecretory cells (*Hamanaka et al., 2010*; *Hewes et al., 2003*; *Park et al., 2008*). Although *dimm* is detected at very low levels and in few cells in our dataset, it showed a bias to cluster 84 (10.7% of cells express *dimm* vs. 0.03% elsewhere). Cluster 84 is also particularly enriched for neuropeptide expression (*Figure 8A and B*) and lacking strong expression of FAN or monoaminergic neuron markers (*Figure 1—figure supplement 7*; *Figure 6C*; *Figure 7B*). They also express several genes involved in neuropeptide processing (see *Figure 8C and D*; *Figure 1—source data 1*). Most neuropeptides are expressed in non-overlapping subsets of the cluster (*Figure 8E*), suggesting that a particular neuropeptide does not drive the unifying identity of the cluster, but rather that it is generically peptidergic. The *Glycoprotein hormone alpha 2* (*Gpa2*) and *Glycoprotein hormone beta*

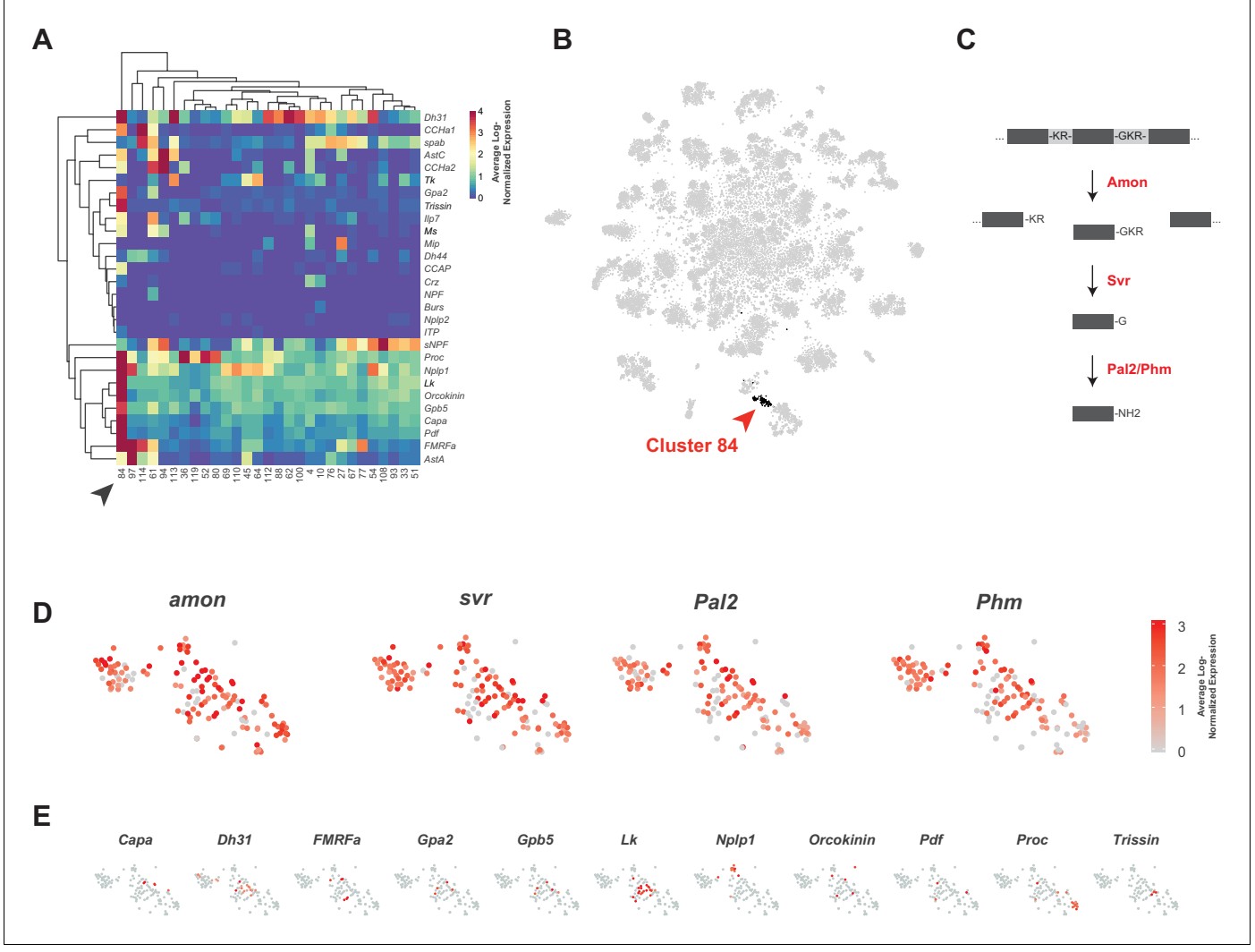

**Figure 8.** Identification of neuropeptide expressing cells. (**A**) Heatmap of the mean log-normalized expression (0–4), by cluster (bottom), of neuropeptide genes (right). Only clusters with at least one neuropeptide gene expressing in excess of mean log-normalized value of 10 are shown. Only neuropeptide genes that were expressed in at least one cell at a level >10 transcripts per cell are shown. Arrowhead highlights cluster 84. Dendrograms represent hierarchical clustering. (**B**) t-SNE plot highlighting neuropeptide expressing cluster 84. (**C**) Schematic of neuropeptide processing steps highlighting enzymes (in red) identified as enriched markers for cluster 84. Propeptides are cleaved by the pro-hormone convertase *Amontillado* (*Amon*); the carboxypeptidase *Silver* (*Svr*) then removes the C-terminal cleavage sequence. C-terminal amidation occurs through the combined actions of *Peptidyl-α-hydroxyglycine-α-amidating lyase 2* (*Pal2*) and *Peptidylglycin-α-hydroxylating monooxygenase* (*Phm*) (reviewed in *Pauls et al., 2014*). (**D**, **E**). Expression of neuropeptide processing enzymes (**D**) and multiple neuropeptide genes (**E**) in t-SNE cluster 84. Intensity of red is proportional to the log-normalized expression level.

The online version of this article includes the following source data and figure supplement(s) for figure 8:

**Source data 1.** Neuropeptide expression levels.
**Figure supplement 1.** Neuropeptide gene expression levels.
**Figure supplement 2.** Analysis of VNC data removing upper limit on transcript number (UMI) per cell.
**Figure supplement 3.** *Orcokinin* expression analysis.

5 (*Gpb5*) genes, whose peptides form heterodimers, are co-expressed in the same cells in this peptidergic cluster (*Sellami et al., 2011*).

## High neuropeptide gene expression reveals important technical considerations

When investigating neuropeptide expressing cluster 84 (NP cluster) we noticed that the median number of transcripts in the cluster was 5277, which was far greater than that observed in the data set as a whole, 2497 (*Figure 8—figure supplement 1A*). This large number of transcripts is due, in part, to high expression levels of the neuropeptide genes themselves. For example, the maximum expression level of *Leucokinin* (*Lk*) was 2629 transcripts in a single cell (*Figure 8—figure supplement 1B*), whereas the maximum expression level of an average gene was 11 transcripts per cell. Amongst neuropeptide genes, *Lk* is not an anomaly, as 18 of the top 40 most highly expressed genes per cell encode neuropeptides (*Figure 8—figure supplement 1C*). All 28 of the expressed neuropeptide genes had a maximum observed expression above the 93[rd] percentile (see methods), with 23 of the 28 above the 99[th] percentile (*Figure 8—source data 1*). In some cases, these genes represented 40% of a cell's captured transcriptional output (*Figure 8—figure supplement 1D*). Similar patterns of high neuropeptide precursor gene expression have also been seen in the mouse cortex (*Smith et al., 2019*).

For all previous data analyses, we used an upper limit of 10,000 total transcripts (nUMI) per cell to remove potential outliers. Given the high expression seen for neuropeptide genes, we repeated our analysis without this cut-off, revealing that many cells in the neuropeptide (NP) and *Proctolin+* motor neuron (*Proc⁺* MN) clusters express more than 10,000 transcripts (*Figure 8—figure supplement 2*). It is worth considering the filtering cut-off when studying cells with high transcriptional output, and the genes expressed therein, whose expression can exceed the threshold. For example, the maximum observed expression of *Orcokinin* was 23,092 transcripts.

Another important consideration when investigating cells expressing genes at very high levels, as many neuropeptide cells do, is the consequence of these cells rupturing during the dissociation process. Once ruptured, large numbers of neuropeptide transcripts could be present in the ambient solution, leading to background levels of neuropeptide transcripts being 'picked up' with non-expressing cells. Our data for the neuropeptide *Orcokinin* illustrates this point (*Figure 8—figure supplement 3*). Orcokinin is expressed in just 5 neurons in the adult VNC, two pairs of neurons in thoracic neuromeres, with additional expression in one unpaired neuron in the abdominal neuromere (*Chen et al., 2015*). In our data, 5 cells in the NP cluster show *Orcokinin* expression at a level of more than 100 transcripts (*Figure 8—figure supplement 3*). However, almost 8000 cells, across all clusters, also appear to express *Orcokinin* at a level of just 1–10 transcripts (*Figure 8—figure supplement 3*). We speculate that the low-level expression outside the NP cluster reflects background levels, due to rupture of *Orcokinin* expressing cells, while the expression above 100 transcripts within the NP cluster is *bona fide*.

## Non-neuronal cell types: Glia

Glia are key regulators of nervous system physiology maintaining the concentration of chemicals in the extracellular environment. We identified glia cells in our VNC data using the established glial markers *reversed polarity* (*repo*) (*Xiong et al., 1994*) and the long non-coding RNA *MRE16* (*Davie et al., 2018*; *Figure 1—figure supplement 7*). Five clusters, representing 3.6% of the total cells, were highly enriched for *repo* and *MRE16*. In addition, these clusters lacked expression of neuronal markers, such as *elav*, *nSyb*, and *noe* (*Figure 1—figure supplement 7*). Two of these clusters contained cells that were positive for both glial markers and cells positive for neuronal markers, suggesting mixed populations. To define specific glial cell types, we performed a sub-clustering analysis of the five glial clusters (*Figure 9A*), while removing any cell with a neuronal signature (see methods). This sub-clustering revealed four distinct glial subpopulations that could be classified based on enriched expression of established markers (*Figure 9B*): *astrocytic leucine-rich repeat molecule* (*alrm*) for astrocytes (*Doherty et al., 2009*), *I'm not dead yet* (*Indy*) for surface glia (*Knauf et al., 2002*), *wrapper* for cortex glia (*Noordermeer et al., 1998*), and *Excitatory amino acid transporter 2* (*Eaat2*) for ensheathing glia (*Stahl et al., 2018*).

## Surface glia

Consistent with the established metabolic role of glia in the brain, many solute carrier (SLC) membrane transporters are enriched in glial cell types (*Figure 9D*; *Figure 9—source data 1*). Surface

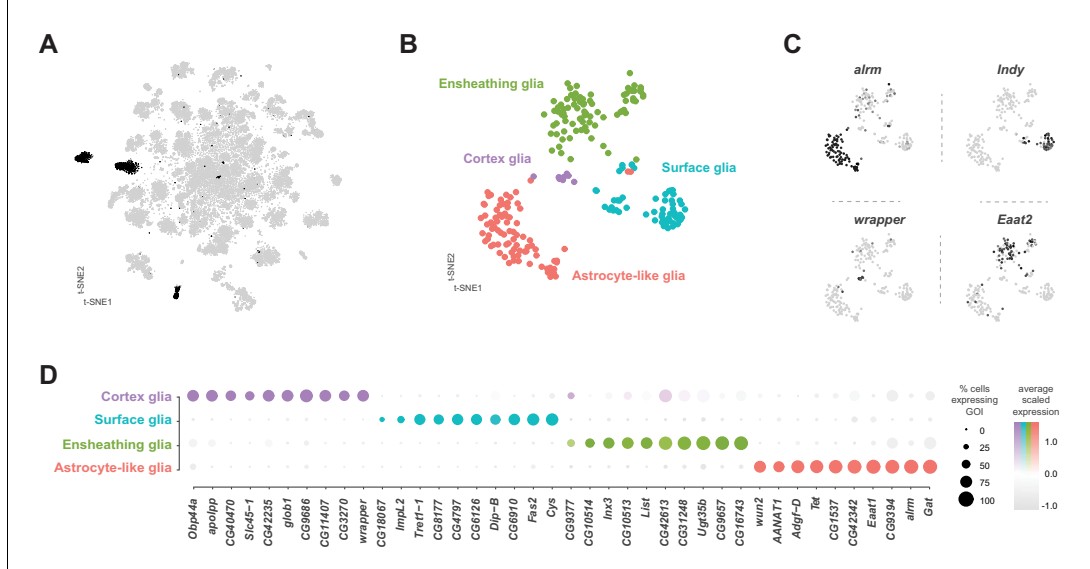

**Figure 9.** Identification and characterization of glial cell sub-types. (**A**) t-SNE plot highlighting glial cell clusters. Clusters 23, 24, 70, 98, and 106 are shown in black, as they were determined to be glia based on differential expression of established glial and neuronal biomarkers (**Figure 1—figure supplement 7**). (**B**) t-SNE plot showing sub-clustering analysis of glial cells. Distinct clusters are color-coded and named. (**C**) Expression of established glial subtype-specific biomarkers used to determine cluster identity. *astrocytic leucine-rich repeat molecule* (*alrm*), for astrocytes; *I'm not dead yet* (*Indy*) for surface glia; *wrapper* for cortex glia; *Excitatory amino acid transporter 2* (*Eaat2*) for ensheathing glia. Expression shown in black, intensity is proportional to the log-normalized expression levels. (**D**) Dot plot of the top 10 genes in each glial sub-type based on fold-enrichment (**Figure 9— source data 1**). Size of dots represent percent of cells in cluster expressing gene of interest (GOI); intensity of color reflects average scaled expression. **Figure 9—source data 1** contain the full list of significantly enriched genes.

The online version of this article includes the following source data and figure supplement(s) for figure 9:

**Source data 1.** List of marker genes for the glial sub-clusters shown in **Figure 9D**.
**Figure supplement 1.** Identification of neuronal and glia clusters.

glia, which form the blood-brain barrier (BBB), metabolically insulate the nervous system from the hemolymph. The BBB also subserves the considerable energetic demands of the nervous system (**Laughlin et al., 1998**), by efficiently transporting sugars, ions, and metabolites between the hemolymph and brain (**Limmer et al., 2014**). Surface glia strongly express the previously described surface glia marker SLC2 *Trehalose transporter 1–1* (*Tret1-1*), which as the name suggests transports trehalose, the main carbohydrate in the hemolymph, into the nervous system (**Volkenhoff et al., 2015**). Surface glia also contain high levels of a putative SLC2 sugar transporter *CG4797*, previously reported to be expressed in the perineural glia of the optic lobe (**Figure 9D**; **Konstantinides et al., 2018**). Both of these sugar transporters are proton-dependent and therefore, efficient sugar transport requires glial cells to have lower $H^+$ concentration than the hemolymph (**Kikuta et al., 2012**). We found enriched expression of multiple V-type ATPase $H^+$ pumping genes (p-value=$3.9 \times 10^{-9}$ based on DAVID analysis; **Figure 9—source data 1**; **Chintapalli et al., 2013**) in surface glia. Expression of the SLC bicarbonate transporter *CG8177*, which has been shown to reduce extracellular pH (**Overend et al., 2016**), and the *Ecdysone-inducible gene L2* (*ImpL2*) were also enriched in surface glia (**Figure 9D**). *ImpL2* antagonizes *insulin-like peptide 2* (*Dilp2*) and inhibits insulin/insulin-like growth factor signaling (**Honegger et al., 2008**) suggesting that the BBB is responsive to the metabolic demands of the fly.

## Astrocytes and ensheathing glia

Over 70% of the VNC astrocyte cell-specific markers are shared with those in astrocytes in the mid-brain (**Croset et al., 2018**). This similarity suggests that there is minimal regional specialization of astrocyte identity and function within the CNS. All astrocytes and ensheathing glia in the thoracic VNC are derived from the same lineages as leg motor neurons (**Enriquez et al., 2018**). However, we did not find evidence of a lineage-specific transcriptional code in these glial cell types (**Figure 9—**

*source data 1*). Astrocytes instead, strongly express the paired-like homeobox TF *CG34367*, previously reported to be expressed in astrocytes throughout development (*Huang et al., 2015*). In contrast, ensheathing glia express the POU homeobox TF *ventral veins lacking* (*vvl*) (*Anderson et al., 1995*). These TFs can, therefore, be considered as novel markers for these adult VNC cell types. The rarity of TF markers is in accord with the finding that the morphological specificity of motor neurons depends on a unique TF code, whereas astrocytes and ensheathing glia show plasticity in their morphologies dependent on their position, rather than a distinct TF code (*Enriquez et al., 2018*).

### Non-neural 'contamination'

We identified two clusters of cells that represent contamination during the dissection process (*Figure 1—figure supplement 7B*). Cluster 99 was determined to be salivary gland tissue, since the cluster markers for cluster 99 contains 8 of the top 10 genes enriched in salivary gland tissue in the FlyAtlas 2 dataset (*Figure 1—source data 1*; *Leader et al., 2018*). Cluster 116 appears to be sperm as many of the markers for this cluster are unique to the testis and sperm (*Figure 1—source data 1*; *Witt et al., 2019*).

## Discussion

As the field tries to categorize nervous systems at higher resolution, the question arises what precisely constitutes a cell type? The goal itself seems straightforward in principle, to find a way to define different groups of cells that carry out distinct tasks. Single-cell mRNA sequencing techniques provide a possible route to answering this question by allowing the neuronal transcriptome of thousands of individual cells within a complex nervous system to be collected in parallel. To this end, we generated a single-cell transcriptional atlas that reveals extensive cellular diversity in the adult *Drosophila* ventral nerve cord. In combination with previous single-cell data from the antennal lobe, optic lobe, and brain (*Croset et al., 2018*; *Davie et al., 2018*; *Konstantinides et al., 2018*; *Li et al., 2017*), our data contribute towards a comprehensive cell atlas representative of the entire adult central nervous system.

Distinguishing between some cell types such as neurons and glia is relatively straightforward, but the extent of neuronal diversity provides a real challenge. Different neuronal types transmit particular neurotransmitters, neuropeptides and monoamines, and they also respond to a variety of these signals using their complement of cell-surface receptors. Specific neurons might also express unique ion-channels and cell-signaling cascades that provide the cell with a range of electrophysiological characteristics, and potential mechanisms of plasticity. In principle, we can also define neuronal cell types by characterizing their neuroanatomy - where the neurons are located and to which neurons they are pre- and post-synaptically connected. Since neurons acquire their anatomy through developmental programs, it was not known whether this information would remain accessible in any form in the snapshot of the transcriptome of mature fully differentiated adult VNC. However, one of the most evident cluster-defining features of our VNC data is the abundance of transcription factors and cell-adhesion molecules that are classically thought of as being developmental.

The cluster-defining TFs are particularly useful for annotating the cell types in the VNC. Decades of work has investigated the development and structural organization of the VNC and has described roles for these genes in developmental specification of hemilineages (*Baek and Mann, 2009*; *Birkholz et al., 2015*; *Bossing et al., 1996*; *Lacin et al., 2019*; *Lacin and Truman, 2016*; *Prokop and Technau, 1991*; *Schmid et al., 1999*; *Shepherd et al., 2016*; *Shepherd et al., 2019*; *Truman and Bate, 1988*; *Truman et al., 2004*). Hemilineages have been proposed to be the functional units of the VNC, representing fundamental organizational principles for connectivity (*Harris et al., 2015*). Our data are entirely consistent with the hemilineages as functional units with each hemilineage made up of a population of neurons that share morphological, transcriptional, and neurochemical features. They represent a familial unit and our data clearly in an unbiased way, pulls them out as separate and identifiable genetic units, reinforcing the idea that the hemilineages are functional groups that share molecular/genetic identity as well as morphology and function. Moreover, we can see that hemilineages are not all homogenous in their composition; there are distinct subtypes evident from the single-cell data. For example, we documented that *kn* and *twz* are expressed in distinct subsets of hemilineage 23B. The fact that subtypes are seen in our data, as well

as the number of subtypes for a given hemilineage, is consistent with what others have observed anatomically (D. Shepherd, pers. comm).

We also found continued expression in hemilineages of neurodevelopment genes, such as *br*, *Imp*, and *pros*, which may signify cell birth order and therefore further distinguish cell types. Expression levels of *pros* indicate the birth order of post-embryonic leg motor neurons belonging to hemilineage 15B, that precedes muscle-specific innervation (*Baek et al., 2013*; *Baek and Mann, 2009*). Relative levels of *Imp* and *pros* in our predicted hemilineage 15B, can therefore be used to infer muscle-specific innervation along the leg. *Imp* and *pros* expression may also distinguish between embryonic born and post-embryonic born neurons. Mature (embryonic) neurons of the larval VNC show elevated *Imp* expression, whereas immature (post-embryonic) neurons have higher levels of *pros*, relative to each other (*Etheredge, 2017*). Dopaminergic neurons in the PAM1 cluster in the larva are the only dopaminergic neurons to be born post-embryonically (*Hartenstein et al., 2017*). So, in some instances, embryonic vs. post-embryonic identity can also be used to identify neurons within a given class.

Fast-acting neurotransmitter identity is acquired in a hemilineage dependent manner, amongst post-embryonic neurons (*Lacin et al., 2019*), but the underlying transcriptional programs appear to be complex (*Estacio-Gómez et al., 2019*). We did not identify single TFs that globally defined FAN identity. Different neurons have been reported to use different TFs to define the same FAN identity in the adult fly optic lobes (*Konstantinides et al., 2018*) and in *C. elegans* (*Hobert and Kratsios, 2019*). In addition, a recent gene-profiling study of cholinergic, glutamatergic, and GABAergic neurons across development found that many FAN-specific TFs are transiently expressed at particular times, and only a few remain constant (*Estacio-Gómez et al., 2019*). As noted previously (*Lacin et al., 2019*), we found some TFs that are restricted to individual FAN-specific clusters. Some GABAergic clusters express *Dbx*, *vg*, *D*; some cholinergic neurons express *unc-4* and some glutamatergic neurons express *ems*. We also found *Lim3* to be expressed in cholinergic and glutamatergic neurons, but not in GABAergic neurons (*Lacin et al., 2019*). This is in contrast to what is seen in the optic lobes, where *Lim3* specifically regulates GABAergic cellular identity (*Konstantinides et al., 2018*). Yet, consistent with the optic lobes, *Lim3* is only expressed in Notch$^{off}$ hemilineages in the VNC (*Li et al., 2017*). Therefore, the TF code specifying FAN identity may change across development and also differ depending on the regional context.

*Hox* gene expression (*Antp*, *Ubx*, *abd-A*, and *Abd-B*) developmentally define the neuromeres of the VNC. Finding that established neuromere- and hemilineage-enriched developmental markers are expressed in the 5 day old adult, suggests they may continue to act in these differentiated cells to maintain the distinct transcriptional profile underlying neuronal identity and function (*Deneris and Hobert, 2014*). It will therefore be interesting to investigate functional consequences of disrupting these potential maintenance programs.

Not all cells group by their developmental lineage, for example, monoaminergic cells formed a distinct cluster, despite originating from multiple different neuroblasts (*Schmid et al., 1999*). We also identified a cluster of neurosecretory cells that includes cells with different developmental origins (*Park et al., 2008*). High expression levels of neuropeptides in this cluster is consistent with features of neurosecretory biology. Neuropeptides are often produced by small numbers of cells, yet they can be broadly released and by volume transmission and/or secretion into the hemolymph act at a distance to modulate disparate neural circuits, physiological functions and behaviors (*Nässel, 2018*). The extraordinarily high-level expression of pre-pro-neuropeptide genes in these cells suggests that producing these molecules presents a considerable burden. The TF *dimmed* (*dimm*), which marks neurosecretory cells of this type, has been shown to promote neurosecretory identity and suppress FAN identity (*Hewes et al., 2003*; *Park et al., 2008*).

In this study, we have demonstrated the predictive power of the single-cell atlas of the VNC. Despite the caveats associated with this technique, such as sparse sampling of a cell's transcriptome and ambient RNA contamination, cellular identity signals, especially developmental programs, are surprisingly robust. Defining cell types based on clustering should be viewed as an exploratory rather than a confirmatory process (*Crow and Gillis, 2019*). It is worth noting that the genes identified using this technique that most robustly define cell clusters do not necessarily reflect their importance for cell type function, however many will no doubt be useful for the generation of tools to study and ultimately define cell phenotypes. Our results suggest many new directions for further investigation. For example, understanding the correspondence and potential causal relationships

between transcriptomic signatures and specific anatomical, physiological and functional properties, and how these relationships change depending on cell state.

## Materials and methods

### Fly strains and husbandry

The fly strain used for VNC analysis (+/w*; UAS-Stinger, 13XLexAop2-IVS-tdTomato.nls/+; dsx$^{GAL4}$/+) was a genetic cross between w*; UAS-Stinger, 13XLexAop2-IVS-tdTomato.nls males and +; +; dsx$^{GAL4}$ virgin females (*Rideout et al., 2010*). All flies were reared at 25°C in a 12:12 hr light: dark cycle on standard food at 40–50% relative humidity. Virgin males and females were collected and stored individually. Flies were aged 5 days post-eclosion at 25°C prior to dissection. Additional fly strains used in this study include: (1) wild-type Canton-S (2) y$^1$w*; Mi{Trojan-GAL4.0}Vmat [MI07680-TG4.0] (BDSC:66806) (3) y$^1$w*; Mi{Trojan-GAL4.1}Lim3[MI03817-TG4.1]/SM6a (BDSC:67450) (4) y$^1$w*; {Mi{Trojan-GAL4.2}kn[MI15480-TG4.2]/SM6a (BDSC:67516) (5) y$^1$w*; Mi{Trojan-GAL4.2}twz[MI14153-TG4.2]/SM6a (BDSC:76758) (6) y$^1$w* Mi{Trojan-GAL4.0}acj6[MI07818-TG4.0]/FM7c (BDSC:77788) (7) w*; {UAS-Stinger} denoted as UAS-nGFP (*Barolo et al., 2000*) (8) w*; UAS-myr-GFP-V5-P2A-H2B-mCherry-HA/TM3, Ser (aka UAS-WM; *Chang et al., 2019*).

### VNC single-cell sample preparation

The VNC dissociation protocol was carried out as described previously (*Croset et al., 2018*). 40 Male and 40 female VNCs (20 per sexed replicate) were individually dissected in toxin-supplemented ice-cold calcium- and magnesium-free DPBS (Gibco, 14190–086 + 50 µM D(−)−2-Amino-5-phospho-novaleric acid, 20 µM 6,7-dinitroquinoxaline-2,3-dione and 0.1 µM tetrodotoxin). Each replicate was then washed in 1 mL ice-cold toxin-supplemented Schneider's medium (tSM: Gibco, 21720–001 + toxins, as above). VNCs were then incubated for 30 min in 0.5 mL of tSM containing 1 mg/mL papain (Sigma, P4762) and 1 mg/mL collagenase I (Sigma, C2674). VNCs were washed once more with tSM and subsequently triturated with flame-rounded 200 µL pipette tips. Dissociated VNCs were resuspended into 1 mL PBS + 0.01% BSA and filtered through a 10 µm CellTrix strainer (Sysmex, 04-0042-2314).

### Data processing

Libraries were made using the Chromium Single Cell 3' v2 kit from 10x Genomics. Cells were loaded in accordance with 10x Genomics documentation, with the aim of 5000–8000 cells per sample. The samples were sequenced with 8 lanes of Illumina HiSeq4000 by Oxford Genomics Centre. We obtained a mean of 20,550 reads per cell and mean sequence saturation of 71%. The fastq data were processed with Drop-seq_tools v2.1.0 (*Macosko et al., 2015*) and aligned to the *D. melanogaster* genome (R6.13) to generate digital expression matrices for each sample.

### Data analysis with seurat

The digital expression matrices were analyzed with the Seurat 2.3.4 R package (*Satija et al., 2015*). Genes expressed in fewer than 3 GEMs (Gel Bead-In EMulsions) were removed. GEMs with more than 10,000 UMI (1.4% of GEMs, 3.6 standard deviations away from the median) were removed as outliers (unless otherwise mentioned). GEMs with fewer than 1,200 UMI were removed. The lower limit was determined using a histogram of the number of UMI per GEM. This distribution has a local minimum between 1000 and 1,200 UMI. GEMs with fewer than 200 genes, or more than 15% mitochondrial derived UMI were removed. This resulted in 26,768 cells in total and 2590, 9060, 6522, 8596 cells for the four replicates, respectively. Sex-specific replicates were merged with the 'Merge-Seurat' function and normalized with the 'NormalizeData' function, using default parameters. The merged sex-specific groups were then scaled with the 'ScaleData' function while regressing out variation due to replicate, nUMI, and proportion mitochondrial transcript. Low expression level, and low dispersion cut-offs of 0.001 were used to identify variable expressed genes in each sex. The intersection of these genes was used to perform a canonical correlation analysis (CCA) with the first 45 dimensions. t-distributed stochastic neighbor embedding (t-SNE) was performed, with perplexity of 30, theta of 0.05, and 20,000 iterations, on the data to reduce the dimensionality to 2 for visualization. Clusters were defined using the 'FindClusters' function with the default Louvain algorithm and

using 45 dimensions and a cluster resolution of 12 (unless otherwise mentioned). Comparison of different cluster resolutions was evaluated with the 'clustree' package (*Zappia and Oshlack, 2018*). Cluster resolutions above 12 yielded few new clusters and increased the between cluster exchanges. Positively enriched cluster markers were identified using the '*FindAllMarkers*' function with a negative binomial distribution test and fold enrichment of at least 0.5 and a Bonferroni adjusted p-value of less than 0.05. Neuronal identity was determined by having an average scaled expression (calculated with the '*AverageExpression*' function in '*Seurat*') of at least 0.15 for any of the following neuronal markers (*elav*, *nSyb*, *para*, *VAChT*, *ChAT*, *Gad1*, *VGAT*, *VGlut*, *noe*). All other clusters were deemed to be non-neuronal, with the exception of neuropeptide-expressing cluster 84. Glial identity was determined in a similar manner with the following glial markers (*repo*, *alrm*, *wrapper*, *Indy*). Functionally related gene enrichment analysis on neuronal cluster markers was performed using DAVID (*Huang et al., 2009*).

## Comparisons to bulk sequence data and between replicates

FlyAtlas 2 bulk RNA-seq data of the adult VNC was obtained from http://flyatlas.gla.ac.uk/FlyAtlas2/index.html (*Leader et al., 2018*). The convert IDs tool from Flybase was used to convert gene symbols and Flybase ID between release versions (http://flybase.org/convert/id). Pseudo-bulk normalized expression from the filtered single-cell VNC data was then compared to the pooled female and male FPKM from the FlyAtlas 2 data set. Pearson correlation coefficients were calculated with the '*stat_cor*' function from the '*ggpubr*' package. Pseudo-bulk normalized expression from each replicate were compared to each other, and correlations were calculated with '*stat_cor*'. Alignment of the replicates was assessed with the '*CalcAlignmentMetric*' function in '*Seurat*'. The '*AverageExpression*' function in '*Seurat*' along with the '*correlate*' function in '*corrr*' were used to determine the correlations of gene expression between the replicates at the cluster level.

## Comparison with brain Single-Cell data

Loom files for the mid-brain (*Croset et al., 2018*) and the whole (*Davie et al., 2018*) were downloaded from 'scope.aertslab.org'. Digital expression matrices of filtered cells were extracted from the loom files and reanalyzed with similar parameters to those used for the VNC. The data were normalized with the '*NormalizeData*' function with default parameters. The data were scaled with the '*ScaleData*' function while regressing out variation due to nUMI and proportion mitochondrial expression. Variably expressed genes were identified with the '*FindVariableGenes*' function with the following cutoffs: x.low.cutoff = 0.001, x.high.cutoff = Inf, y.cutoff = 0.001. A principal component analysis was performed using the identified variable genes. Clusters were identified with the '*FindClusters*' function using the default Louvain algorithm. A cluster resolution of 2.5 and the first 50 PCAs were used for the mid-brain data (*Croset et al., 2018*), and a cluster resolution of 2 and the first 82 PCAs were used for the brain data (*Davie et al., 2018*). t-SNE was performed with perplexity of 30, theta of 0.1, and 20,000 iterations, on the data to reduce the dimensionality to two for visualization. Significantly enriched genes for each cluster were identified using the '*FindAllMarkers*' function using the negative binomial test with a log fold-change threshold of 0.5 while repressing out variation due to replicate, and a Bonferroni adjusted p-value of less than 0.05. The resulting cluster markers for the mid-brain and brain data sets were compared to the VNC data cluster markers (excluding the salivary gland cluster and the sperm cluster). The overlap of markers was visualized with an Euler plot using the 'eulerr' package in R.

## Predicting hemilineage identity

The average expression of previously established hemilineage markers (*Lacin et al., 2019*; *Venkatasubramanian and Mann, 2019*) across all neuronal clusters was calculated with the '*AverageExpression*' function in '*Seurat*'. Predicted hemilineage identities were deduced from these expression patterns. With these assignments, we used the '*FindAllMarkers*' function (with the above settings) to identify novel biomarkers for each predicted hemilineage. Hemilineage specific cell counts (inferred from *Birkholz et al., 2015* and *Lacin et al., 2019*) were compared to the cell counts from our predicted labeled hemilineages. To approximate post-embryonic thoracic cells, only *pros*[+], *abd-A*[-], *Abd-B*[-] cells were considered.

### Assigning fast-acting neurotransmitter identity

The following genes were used to assign fast-acting neurotransmitter (FAN) identity; *Vesicular acetylcholine transporter* (*VAChT*) and *Choline acetyltransferase* (*ChAT*) for acetylcholine, *Vesicular glutamate transporter* (*VGlut*) for glutamate, and *Vesicular GABA Transporter* (*VGAT*) and *Glutamic acid decarboxylase 1* (*Gad1*) for GABA. Cells from clusters with average scaled expression above 0.5 for *Gad1* or *VGAT*, and average scaled expression less than 0 for the other markers were all assigned to be GABAergic. Glutamatergic and cholinergic cells were assigned similarly. For clusters with an average expression less than 0.5 for all markers, cells were assigned FAN identity at a cell-by-cell level. For these remaining cells, we established different criteria for assigning cholinergic vs. glutamatergic or GABAergic since the cholinergic markers were expressed at much lower levels. If both *VAChT* and *ChAT* had a log-normalized expression greater than 0, then the cells were assigned to be cholinergic. Additionally, if either *VAChT* or *ChAT* had a log-normalized expression greater than 2, the cells were assigned to be cholinergic. Otherwise, if *VGlut* had a log-normalized expression greater than 2 and greater than either *VAChT* or *Gad1*, then the cells were assigned to be glutamatergic. Otherwise, if *Gad1* had a log-normalized expression greater than 2 and greater than either *VAChT* or *VGlut*, the cells were assigned to be GABAergic. All remaining cells were labelled undefined.

### Sub-clustering of monoaminergic cells

*Vmat* expressing cells from clusters 72 and 84 were sub-clustered. Identification of variable genes and canonical correlation analysis were performed as above. t-SNE was performed using the first 7 dimensions and a cluster resolution of 1.2 was used to identify clusters. Positively enriched cluster markers were identified as above.

### Sub-clustering of glial cells

Clusters 23, 24, 70, 98, and 106 were identified as being predominantly glial cells. Cells from these clusters were sub-clustered. Any cell expressing *elav*, *nSyb*, *VAChT*, *VGlut*, or *Gad1* was removed. Identification of variable genes and canonical correlation analysis were performed as above. t-SNE was performed using the first 6 dimensions and a cluster resolution of 0.9 was used to identify clusters. Positively enriched cluster markers were identified as above.

### R packages and plotting

Expression heatmaps were drawn using the 'pheatmap' package, correlation heatmaps were drawn using the 'ggcorplot' package. Chord diagrams were drawn using the 'circlize' package (*Gu et al., 2014*), and represent the relationship between cluster markers and the clusters in which they are significantly enriched, the data is taken from *Figure 1—source data 1*. No weighted relationships are inferred in chord diagrams. Scatter plots and bar charts were drawn using ggplot2 in R. A full list of packages used, with their version numbers can be found at https://github.com/aaron-allen/VNC_scRNAseq/blob/master/sessionInfo.txt. All plots were edited in Adobe Illustrator.

### Phylogeny

The multiple sequence alignment (not shown) and the phylogenetic tree were created with Clustal Omega (*Sievers and Higgins, 2018*) and FigTreeV1.4.4 (http://tree.bio.ed.ac.uk/software/figtree/), respectively.

### Immunohistochemistry

Flies were reared at 25°C and aged for 5 days prior to dissection and staining as per *Rideout et al. (2010)* with the following modifications: VNC samples were fixed in 4% PFA (40 mins at room temperature) immediately following dissection, to maintain tissue integrity and minimize cell loss. Samples were pre-incubated in 5% NGS overnight at 4°C. Samples were then incubated with primary antibodies for 3 days at 4°C, followed by an overnight incubation in secondary antibodies at 4°C. Primary antibodies used were: mouse mAb nC82 (1:50), mouse anti-AbdA (1:100, C-11), mouse anti-AbdB (1:10, 1A2E9-s), mouse anti-Acj6-s (1:20), mouse anti-AntP (1:20, 8C11-s), mouse anti-Ubx (1:20, FP3.38-s), rat anti-CadN (1:30, DN-Ex #8;) from DSHB, Univ. of Iowa. Additional primary antibodies used were chicken anti-GFP (1:1200, Abcam) and anti-AbdA (1:100, C-11 Santa Cruz

Biotechnology). Secondary antibodies used included: anti-chicken Alexa Fluor488, anti-mouse Alexa Fluor633, anti-rat Alexa Fluor546, (1:300, Invitrogen Molecular Probes, Carlsbad, CA). Samples were left in 70% Glycerol/30% PBT overnight at 4°C prior to mounting with Vectashield (Vector Labs) and imaged with a Leica SP5 Microscope. Stacks of optical sections were generated at 0.5 μm intervals. Images were processed in Imaris 8.2.1 (Bitplane Scientific, AG, Zürich).

## Data and code availability
Raw sequencing files (fastq) and digital expression matrices from each replicate are available from the Gene Expression Omnibus (GSE141807). Code used in this analysis is available from GitHub (https://github.com/aaron-allen/VNC_scRNAseq; *Allen, 2020*; copy archived at https://github.com/elifesciences-publications/VNC_scRNAseq).

## Acknowledgements
We thank Tetsuya Nojima and Annika Rings for dissection assistance. We thank David Shepherd, Devika Agarwal, Deniz Erezyilmaz, Julian Dow, and members of the Goodwin lab for helpful discussions and critical reading of the manuscript. We thank the Bloomington Stock Center for flies. We thank Josh Dubnau for the *UAS-WM* line. Sequencing was carried out by The Oxford Genomics Centre. We thank Deborah J Andrew and Caitlin D Hanlon for assistance generating a Figure of the GPCRs, and Andreas Prokop for the adult *Drosophila* illustration. SW is supported by a Wellcome Principal Research Fellowship (200846/Z/16/Z) and ERC Advanced Grant (789274). This work was funded by a Wellcome Investigator Award (106189/Z/14/Z) to SFG, and a Wellcome Collaborative Award (209235/Z/17/Z) to SFG and SW.

## Additional information

### Funding

| Funder | Grant reference number | Author |
| --- | --- | --- |
| Wellcome Trust | Wellcome Trust Investigator Award (106189/Z/14/Z) | Aaron M Allen<br>Megan C Neville<br>Stephen F Goodwin |
| ERC Advanced Grant | 789274 | Vincent Croset<br>Christoph Daniel Treiber<br>Scott Waddell |
| Wellcome Trust | Wellcome Principal Research fellowship (200846/Z/16/Z) | Scott Waddell |
| Wellcome Trust | Wellcome Collaborative Award (209235/Z/17/Z) | Scott Waddell<br>Stephen F Goodwin |
| MRC | Graduate Student Fellowship | Sebastian Birtles |

The funders had no role in study design, data collection and interpretation, or the decision to submit the work for publication.

### Author contributions
Aaron M Allen, Conceptualization, Data curation, Formal analysis, Validation, Investigation, Visualization, Methodology, Writing - original draft, Writing - review and editing; Megan C Neville, Conceptualization, Formal analysis, Validation, Investigation, Visualization, Methodology, Writing - original draft, Writing - review and editing; Sebastian Birtles, Formal analysis, Validation, Investigation, Visualization, Methodology, Writing - original draft, Writing - review and editing; Vincent Croset, Methodology, Writing - review and editing; Christoph Daniel Treiber, Methodology; Scott Waddell, Conceptualization, Supervision, Funding acquisition, Writing - review and editing; Stephen F Goodwin, Conceptualization, Supervision, Funding acquisition, Writing - original draft, Project administration, Writing - review and editing

## Author ORCIDs

Aaron M Allen [iD] https://orcid.org/0000-0002-7961-4392
Megan C Neville [iD] https://orcid.org/0000-0001-8506-9944
Sebastian Birtles [iD] https://orcid.org/0000-0003-4617-4092
Christoph Daniel Treiber [iD] http://orcid.org/0000-0002-6994-091X
Scott Waddell [iD] http://orcid.org/0000-0003-4503-6229
Stephen F Goodwin [iD] https://orcid.org/0000-0002-0552-4140

## Decision letter and Author response

Decision letter https://doi.org/10.7554/eLife.54074.sa1
Author response https://doi.org/10.7554/eLife.54074.sa2

## Additional files

### Supplementary files

• Transparent reporting form

### Data availability

Sequencing data have been deposited in GEO under accession code GSE141807.

The following dataset was generated:

| Author(s) | Year | Dataset title | Dataset URL | Database and Identifier |
|---|---|---|---|---|
| Allen AM, Neville MC, Birtles S, Croset V, Treiber C, Waddell S, Goodwin SF | 2019 | A single-cell transcriptomic atlas of the adult Drosophila ventral nerve cord | https://www.ncbi.nlm.nih.gov/geo/query/acc.cgi?acc=GSE141807 | NCBI Gene Expression Omnibus, GSE141807 |

The following previously published datasets were used:

| Author(s) | Year | Dataset title | Dataset URL | Database and Identifier |
|---|---|---|---|---|
| Leader DP, Krause SA, Pandit A, Davies SA, Dow JAT | 2018 | FlyAtlas 2: a new version of the Drosophila melanogaster expression atlas with RNA-Seq, miRNA-Seq and sex-specific data | https://www.omicsdi.org/dataset/omics_ena_project/PRJEB22205 | Omnics DI, PRJEB22205 |

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
