## [Decision Letter]

**Acceptance summary:**

We all agree that this work provides a timely transcriptomics dataset that will be of interest to a large number of scientists, and that you and your coauthors did an excellent job with the validation and analysis of these data.

**Decision letter after peer review:**

Thank you for submitting your article "A single-cell transcriptomic atlas of the adult *Drosophila* ventral nerve cord" for consideration by *eLife*. Your article has been reviewed by three peer reviewers, and the evaluation has been overseen by a Reviewing Editor and K VijayRaghavan as the Senior Editor. The following individual involved in review of your submission has agreed to reveal their identity: Peter A Sims (Reviewer #2).

The reviewers have discussed the reviews with one another and the Reviewing Editor has drafted this decision to help you prepare a revised submission.

We all agreed that the manuscript is a timely contribution to the *Drosophila* neuroscience community, with likely implications beyond the fly field. In particular, the authors perform single-cell RNA-seq on ~26k cells from the adult *Drosophila* ventral nerve cord (VNC). They use clustering analysis to group these cells into 120 distinct clusters, (110 neuronal, 10 non-neuronal). They compare their clusters to known organizing features of the VNC, including developmental lineage, transcription factor enrichment, receptor expression, neurotransmitter identity, and neuropeptide release. Additionally, the authors use Hox gene expression to try to infer spatial origins of sequenced cells. Overall, this paper provides a useful gene expression resource for fly VNC enthusiasts. There are also a handful of interesting biological insights. Altogether, the single cell sequencing approach reveals a new level of organization of the fly VNC and will therefore be of interest to many labs around the world.

However, based on the reviewers' comments, some revisions are required before *eLife* can offer publication.

Essential revisions:

1) The authors claim to be able to discern specific identifying features about single cells in their dataset, such as spatial distribution (i.e., neuromere identity) and lineage. However, there is no quantitative validation of these claims, only qualitative assessment of a few examples. For ex-ample, the Abstract states, "Cells could also be assigned to specific neuromeres using adult Hox gene expression." Is this really true? Apart from the fact that there is no ground-truth to test this claim from RNA-Seq, the antibody staining of Hox genes is far from "neuromere specific". While there may be a loose relationship between gene expression and spatial location, it does not ap-pear to be sufficient to differentiate between meso and metathoracic with a high degree of certainty. The same pattern seems to be true for lineage identity – in a handful of cases examined in this paper, one can recognize lineages from gene expression. But this is not systematically explored. Readers deserve a more objective sense of what they can infer about single cells from this gene expression dataset. In the absence of a statistical analysis that achieves this, the authors should moderate the claims of the paper and include a more clear discussion about what exactly can be learned about single cells by looking at their gene expression using these methods.

2) The authors state that 28% of cells in their dataset express more than one marker of neuro-transmitter identity, and that this may represent a technical artifact of the single-cell method. This point needs further clarification and discussion. Does this represent contamination? If so, does this provide a means of estimating contamination levels across the entire dataset, assuming that co-expression is not "real"? In addition to citing the antibody staining from Lacin et al., 2019, the authors should use Davis et al., 2020, and Shih et al., 2019, as a points of comparison. Using an alternate technique (TAPIN-seq) that purifies identifies cell-types, these studies find no evidence for co-release in the visual system or the mushroom body.

3) One of the more interesting aspects of the paper is the distribution of neuropeptide-related genes and their remarkably high expression levels. The authors should add a ranked list of npp genes, expression levels, and cognate np-gpcr genes. A similar table is present in the recent paper from Smith et al., 2019. It would also be valuable for the authors to compare their results to those in this recent paper.

4) The authors should more quantitatively compare this dataset to that from the fly brain. Other than transmitter distribution, are there any other major differences? If so, how might these be related to differences in methodology (e.g., drop-seq vs. 10x).

5) The data are presented very much as a resource and the manuscript does not focus on a scientific question or hypotheses. Although the data presented are interesting, relevant, and compelling enough that this should not preclude publication, the authors should add as many conclusions that go beyond this resource perspective as possible. For example, do their data hint at any functional organization of the VNC (e.g. support for the hypothesis that hemilineages are functional units), and can they discuss this a bit more? Are there any other hypotheses for functional units that emerge? How does the amount of cellular diversity compare to the brain? To other organisms? Does their atlas provide any clues as to why there is so much diversity?

6) The main deliverable of these analyses is the clusters they establish in Figure 1 and then use as an organizing scaffold for their more detailed analyses. Clear delineation of these clusters therefore seems very important. I find it vague in the manuscript, however, how the clusters may actually be defined. The authors say that most of the clusters can be identified by unique combinations of marker genes, and Figure 1—figure supplement 1 claims to list the "combinations of genes [that] uniquely identify each cluster". But in that figure there are multiple clusters where the same combination of marker genes is listed (e.g. clusters 14, 33, 34, and 78 all list bi and VAChT as the only marker genes defining the cluster). Is that because these clusters are part of a larger "super cluster" (I note 14, 33, 34, and 78 co-localize in their t-SNE plot)? And is the "super cluster" the level at which gene marker combinations are unique? I think articulating the clustering clearly is critical to the paper because that is the level that future scientists will try to compare to morphological cell types. It would also be helpful if there were a large summary table of their many analyses indicating, for each cluster, which lineage, FAN, monamines, peptides, and Hox genes corresponded.

7) Many of the analyses are illustrated with "Chord diagrams". It's not clear what the reader is supposed to take from these diagrams. Are they showing clustering? Are they showing distribution? A quantification of what we are supposed to take from these would be helpful.

8) The authors have four distinguishable replicates in their data set that were each generated by pooling 20 VNCs. However, for essentially all of the analysis, the data from these replicates are pooled and there is a general lack of assessment of consistency across replicates. Figure 1—figure supplement 3A shows the proportional contribution of each replicate to each cluster, but it is impossible to assess any differences in the composition of each replicate from this plot because the replicates are not sub-sampled to the same cell numbers. For example, Replicate 1 contributes only ~5-10% of the total data. Ideally, one would cluster each replicate separately and determine the extent to which the clusters agree with each other across replicates. At the very least, there should be an assessment of the correlation between cluster compositions across replicates based on the current analysis. In addition, the authors should show a tSNE plot where the cells are colored by replicate (rather than by cluster). This should be done for all of the sub-clustering analysis as well.

9) What is the specific rationale for removing cells with fewer than 1,200 UMIs or more than 10,000 UMIs? The authors state that they are trying to remove multiplets with the 10,000 UMI cutoff. Is there evidence from the data that these barcodes are associated with multiplets (e.g. co-expression known mutually exclusive markers)? In general, this approach is somewhat problematic and arbitrary (e.g. see Stoeckius et al., Genome Biology, 2019 for systematic analysis showing the complications of using coverage to identify multiplets). Within this study, Figure 8 shows one example of how this type of cutoff can bias the inferred cellular composition within this dataset.

10) It is unclear from the Materials and methods section how the clustering was done. The authors mention that "clusters were defined by a shared nearest neighbor method", but shared nearest neighbors is not a clustering algorithm. This is just an approach to constructing a graph from the data that can be used as input to a clustering algorithm. Was this graph used for Louvain community detection or something similar?

11) The authors identified a large number of clusters (120). Did all of these clusters have significant enrichment of distinct marker genes (e.g. based on the negative binomial test that the authors used)? What is the range in the number of cluster-specific markers associated with these clusters? Were certain clusters collapsed based on some criteria?

12) The authors state that they observed a median of 5 HD TFs per cell and 52 TFs in total per cell. These numbers may in large part reflect the indeterminate drop-out rate of the Chromium chemistry. It might be more meaningful to (additionally) report these numbers at the cluster level and also to state how many cluster-specific HDs or HD combinations were observed.

13) The single-cell expression correlation analysis shown in Figure 3B (and throughout) is complicated by transcript drop-out. The scatter plots in Figure 6—figure supplement 1D highlight the difficulty of assessing expression correlation at the single cell level from these data. Three different scatter plots that are quite visually similar are shown, but with very different correlation values that seem to be dependent on cell numbers. My guess is that cell numbers and expression levels significantly impact the correlation values at the single cell level. It would be interesting to check if the reported trends are consistent at the cluster level (e.g. by taking the average expression of all cells in each cluster and computing the correlation coefficients across clusters rather than across cells). It would also be worth doing this calculation separately for different replicates to determine their consistency, since this could be influenced by technical and coverage differences between replicates. This comment is not meant to challenge the overall validity of the authors' claims, which in some cases are orthogonally validated by immunostaining.

14) The analysis in Figure 4 is very compelling. There are many cells in the center of the tSNE plot that do not appear to correspond to any of the known lineage markers assessed in Figure 4 (in comparison to those in the periphery of the plot). These cells in the center are generally less tightly clustered. The authors specifically mention a large unc-4 expressing cholinergic cluster at the center of the plot, but Figure 4—figure supplement 1 shows that this represents a relatively small fraction of the cells at the center. Is this phenomenon due to lower coverage cells clustering at the center of the tSNE plot (it is hard to tell from Figure 1—figure supplement 3C) or are these cells just biologically less distinct or require larger cell numbers to have sufficient coverage?

[Editors' note: further revisions were suggested prior to acceptance, as described below.]

Thank you for resubmitting your work entitled "A single-cell transcriptomic atlas of the adult *Drosophila* ventral nerve cord" for further consideration by *eLife*. Your revised article has been evaluated by K VijayRaghavan as the Senior Editor, a guest Reviewing Editor and three peer reviewers.

The manuscript has been greatly improved. There is one remaining issue that should be addressed before publication:

One reviewer was not convinced by Figure 5F that the cluster markers are segregated, as there seems to be co-labeled cell bodies across both clusters 23B and 8B. This is the only example where antibody staining or anatomical verification was used to confirm the accuracy of the cluster markers. I think it is worth quantifying the extent to which cells are co-labeled or adding labels to the images that make the result more clear.

And:

Figure 1—figure supplement 3B: the label wrong in A, should be "Res. 1"

---

## [Author Response]

Essential revisions:1) The authors claim to be able to discern specific identifying features about single cells in their dataset, such as spatial distribution (i.e., neuromere identity) and lineage. However, there is no quantitative validation of these claims, only qualitative assessment of a few examples. For ex-ample, the Abstract states, "Cells could also be assigned to specific neuromeres using adult Hox gene expression." Is this really true? Apart from the fact that there is no ground-truth to test this claim from RNA-Seq, the antibody staining of Hox genes is far from "neuromere specific". While there may be a loose relationship between gene expression and spatial location, it does not ap-pear to be sufficient to differentiate between meso and metathoracic with a high degree of certainty. The same pattern seems to be true for lineage identity – in a handful of cases examined in this paper, one can recognize lineages from gene expression. But this is not systematically explored. Readers deserve a more objective sense of what they can infer about single cells from this gene expression dataset. In the absence of a statistical analysis that achieves this, the authors should moderate the claims of the paper and include a more clear discussion about what exactly can be learned about single cells by looking at their gene expression using these methods.

It wasn’t our intention, but we do recognize that we have overstated our claims regarding the assignment of cells to specific neuromeres and hemilineages and have moderated our claims throughout the manuscript. We have added a more balanced assessment of the use of single-cell data in the Discussion: “Despite the caveats associated with this technique, such as sparse sampling of a cell’s transcriptome and ambient RNA contamination, cellular identity signals, especially developmental programs, are surprisingly robust. […] It is worth noting that the genes identified using this technique that most robustly defines cell clusters do not necessarily reflect their importance for cell type function, however many will no doubt be useful for the generation of tools to study and ultimately define cell phenotypes.”

2) The authors state that 28% of cells in their dataset express more than one marker of neuro-transmitter identity, and that this may represent a technical artifact of the single-cell method. This point needs further clarification and discussion. Does this represent contamination? If so, does this provide a means of estimating contamination levels across the entire dataset, assuming that co-expression is not "real"? In addition to citing the antibody staining from Lacin et al., 2019, the authors should use Davis et al., 2020, and Shih et al., 2019, as a points of comparison. Using an alternate technique (TAPIN-seq) that purifies identifies cell-types, these studies find no evidence for co-release in the visual system or the mushroom body.

Contamination is a complicated issue which varies on a gene by gene basis depending on expression levels. This is a feature of all single-cell data sets. We recognize that these genes in our dataset might be a good testbed for examining contamination issues, and we are currently investigating this question with our collaborators who specialize in bioinformatic analyses. We predominantly focused on FAN expression at the cluster level to circumvent this issue. We have adjusted this section to read “Co-expression in the VNC may therefore represent contamination due to ambient RNA present in the cell suspension.”. Ambient RNA is discussed further in the neuropeptide section of the Results. We have added an additional sentence to the text to be included with the antibody staining from Lacin et al., 2019: “Nuclear transcriptomic profiles of neuronal subtypes from both the visual system and the mushroom body found no evidence for FAN co-release (Davis et al., 2020; Shih et al., 2019).”

3) One of the more interesting aspects of the paper is the distribution of neuropeptide-related genes and their remarkably high expression levels. The authors should add a ranked list of npp genes, expression levels, and cognate np-gpcr genes. A similar table is present in the recent paper from Smith et al., 2019. It would also be valuable for the authors to compare their results to those in this recent paper.

We have added a table (Figure 8—source data 1) listing the expressed neuropeptide precursor genes; their maximum observed log-normalized expression, the rank of this expression level, the percentile of this expression level, and their associated receptors. We have discussed this data in relation to Smith et al., 2019, and added the citation.

4) The authors should more quantitatively compare this dataset to that from the fly brain. Other than transmitter distribution, are there any other major differences? If so, how might these be related to differences in methodology (e.g., drop-seq vs. 10x).

To compare these VNC data with the existing Drop-seq and 10x data sets, we first re-analyzed the brain data sets using the same statistical test (negative binomial) and cutoffs (avg_logFC>0.5, p_val_adj<0.05) as used in our data set. We compared the overlap in the cluster, defining significantly enriched genes between these data sets. Both 10x data sets generated more cluster markers than the Drop-seq data. There was extensive overlap seen between all data sets. We have added Figure 2—figure supplement 1, Figure 2—source data 1, and Figure 2—source data 2 to illustrate these patterns and discuss them in the text.

5) The data are presented very much as a resource and the manuscript does not focus on a scientific question or hypotheses. Although the data presented are interesting, relevant, and compelling enough that this should not preclude publication, the authors should add as many conclusions that go beyond this resource perspective as possible. For example, do their data hint at any functional organization of the VNC (e.g. support for the hypothesis that hemilineages are functional units), and can they discuss this a bit more? Are there any other hypotheses for functional units that emerge? How does the amount of cellular diversity compare to the brain? To other organisms? Does their atlas provide any clues as to why there is so much diversity?

We thank the reviewers for this comment as it helped us rethink our discussion of hemilineages in the paper, we have now included the following: “Our data are entirely consistent with the hemilineages as functional units with each hemilineage made up of a population of neurons that share morphological, transcriptional and neurochemical features. They represent a familial unit and our data clearly and in an unbiased way, pulls them out as separate and identifiable genetic units, reinforcing the idea that the hemilineages are functional groups that share molecular/genetic identity as well as morphology and function. Moreover, we can see that hemilineages are not all homogenous in their composition; there are distinct subtypes evident from the single-cell data. For example, we documented that kn and twz are expressed in distinct subsets of hemilineage 23B. The fact that subtypes are seen in our data, as well as the number of subtypes for a given hemilineage, is consistent with what others see anatomically (D. Shepherd, pers. comm).”

6) The main deliverable of these analyses is the clusters they establish in Figure 1 and then use as an organizing scaffold for their more detailed analyses. Clear delineation of these clusters therefore seems very important. I find it vague in the manuscript, however, how the clusters may actually be defined. The authors say that most of the clusters can be identified by unique combinations of marker genes, and Figure 1—figure supplement 1 claims to list the "combinations of genes [that] uniquely identify each cluster". But in that figure there are multiple clusters where the same combination of marker genes is listed (e.g. clusters 14, 33, 34, and 78 all list bi and VAChT as the only marker genes defining the cluster). Is that because these clusters are part of a larger "super cluster" (I note 14, 33, 34, and 78 co-localize in their t-SNE plot)? And is the "super cluster" the level at which gene marker combinations are unique? I think articulating the clustering clearly is critical to the paper because that is the level that future scientists will try to compare to morphological cell types. It would also be helpful if there were a large summary table of their many analyses indicating, for each cluster, which lineage, FAN, monamines, peptides, and Hox genes corresponded.

We apologize that the genes listed in the previous Figure 1—figure supplement 1 (now Figure 1—figure supplement 4) were misleading; these have now been removed. The unique combinations of significantly enriched genes are now clearly referenced to Figure 1—source data 1. We have included a new summary table of defined clusters, Figure 1—source data 2, which includes cluster: cell number, number of markers, number of unique markers, predicted FAN/monoamine/peptidergic identity, predicated hemilineage assignment, enriched peptides and enriched Hox gene expression.

7) Many of the analyses are illustrated with "Chord diagrams". It's not clear what the reader is supposed to take from these diagrams. Are they showing clustering? Are they showing distribution? A quantification of what we are supposed to take from these would be helpful.

Chord diagrams represent the relationship between cluster markers and the clusters in which they are significantly enriched; they are used purely as a visualization aid to the data in Figure 1—source data 1. No weighted relationships are inferred. We have added a description to the Materials and methods section of the paper and have more clearly defined their use in the appropriate figure legends.

8) The authors have four distinguishable replicates in their data set that were each generated by pooling 20 VNCs. However, for essentially all of the analysis, the data from these replicates are pooled and there is a general lack of assessment of consistency across replicates. Figure 1—figure supplement 3A shows the proportional contribution of each replicate to each cluster, but it is impossible to assess any differences in the composition of each replicate from this plot because the replicates are not sub-sampled to the same cell numbers. For example, Replicate 1 contributes only ~5-10% of the total data. Ideally, one would cluster each replicate separately and determine the extent to which the clusters agree with each other across replicates. At the very least, there should be an assessment of the correlation between cluster compositions across replicates based on the current analysis. In addition, the authors should show a tSNE plot where the cells are colored by replicate (rather than by cluster). This should be done for all of the sub-clustering analysis as well.

To verify the consistency between replicates, we have added multiple new analyses and figures, and discuss these in the main text. In Figure 1—figure supplement 2, we compared the pseudo-bulk expression profiles of each replicate to each other. All pairwise comparisons have a coefficient of correlation of 0.93 to 0.96. In Figure 1—figure supplement 5B, we have added t-SNE plots highlighting the cell from each replicate. Figure 1—figure supplement 5C, shows a heatmap of all pairwise correlations of expression of every gene separated by cluster. Similarly, we have added t-SNE plots and correlation heatmaps for both the monoamine sub-clustering (Figure 7—figure supplement 1) and the glial sub-clustering (Figure 9—figure supplement 1).

9) What is the specific rationale for removing cells with fewer than 1,200 UMIs or more than 10,000 UMIs? The authors state that they are trying to remove multiplets with the 10,000 UMI cutoff. Is there evidence from the data that these barcodes are associated with multiplets (e.g. co-expression known mutually exclusive markers)? In general, this approach is somewhat problematic and arbitrary (e.g. see Stoeckius et al., Genome Biology, 2019 for systematic analysis showing the complications of using coverage to identify multiplets). Within this study, Figure 8 shows one example of how this type of cutoff can bias the inferred cellular composition within this dataset.

We have added clarification in the Materials and methods section describing how we arrived at these cutoffs, and new figures to illustrate. For the high cutoff of 10,000 UMI we reasoned that these were outliers (3.6 sd away from the median, and only accounting for 1.4% of the raw data). This can be seen in Figure 1—figure supplement 1A. The lower cutoff of 1,200 UMI was determined using a histogram of the number of UMI per cell (Figure 1—figure supplement 1B). There is a local minimum between 1,000 and 1,200 UMI. We agree that a high cutoff for UMI is problematic as a means to remove multiplets. We have changed the phrasing of ‘doublets’ to ‘outliers’.

10) It is unclear from the Materials and methods section how the clustering was done. The authors mention that "clusters were defined by a shared nearest neighbor method", but shared nearest neighbors is not a clustering algorithm. This is just an approach to constructing a graph from the data that can be used as input to a clustering algorithm. Was this graph used for Louvain community detection or something similar?

We apologize, this has now been changed in the Materials and methods section, it states “Clusters were defined using the *‘FindClusters’* function with the default Louvain algorithm and using 45 dimensions and a cluster resolution of 12”.

11) The authors identified a large number of clusters (120). Did all of these clusters have significant enrichment of distinct marker genes (e.g. based on the negative binomial test that the authors used)? What is the range in the number of cluster-specific markers associated with these clusters? Were certain clusters collapsed based on some criteria?

All clusters did have enrichment of distinct sets of marker genes based on the negative binomial test; we have made the reference to this data clearer in the text (see point 6 above). We have made this information more accessible by including information in Figure 1—source data 2 regarding the number of significant markers in each cluster as well as the number of unique markers per cluster. No clusters were collapsed.

12) The authors state that they observed a median of 5 HD TFs per cell and 52 TFs in total per cell. These numbers may in large part reflect the indeterminate drop-out rate of the Chromium chemistry. It might be more meaningful to (additionally) report these numbers at the cluster level and also to state how many cluster-specific HDs or HD combinations were observed.

We have generated new plots (Figure 2—figure supplement 6) that include both the single-cell level and the cluster level counts. These new plots also include a threshold series for the minimum number of cells in the cluster with detectable expression to count the cluster as being positive for that gene. The plots also include a threshold for the number of UMI detected within a cell to count as bona fide expression. We believe these new plots more accurately represent the data.

13) The single-cell expression correlation analysis shown in Figure 3B (and throughout) is complicated by transcript drop-out. The scatter plots in Figure 6—figure supplement 1D highlight the difficulty of assessing expression correlation at the single cell level from these data. Three different scatter plots that are quite visually similar are shown, but with very different correlation values that seem to be dependent on cell numbers. My guess is that cell numbers and expression levels significantly impact the correlation values at the single cell level. It would be interesting to check if the reported trends are consistent at the cluster level (e.g. by taking the average expression of all cells in each cluster and computing the correlation coefficients across clusters rather than across cells). It would also be worth doing this calculation separately for different replicates to determine their consistency, since this could be influenced by technical and coverage differences between replicates. This comment is not meant to challenge the overall validity of the authors' claims, which in some cases are orthogonally validated by immunostaining.

Assessing the correlation of expression at the cluster level did not alter the overall patterns seen at the single cell level (for Figure 3B; Figure 4—figure supplement 2D-F; Figure 6—figure supplement 1A, D). These patterns are also consistent between replicates, at both the single cell and cluster levels of expression. We have added statements to this effect in both the figure legends and the main text.

14) The analysis in Figure 4 is very compelling. There are many cells in the center of the tSNE plot that do not appear to correspond to any of the known lineage markers assessed in Figure 4 (in comparison to those in the periphery of the plot). These cells in the center are generally less tightly clustered. The authors specifically mention a large unc-4 expressing cholinergic cluster at the center of the plot, but Figure 4—figure supplement 1 shows that this represents a relatively small fraction of the cells at the center. Is this phenomenon due to lower coverage cells clustering at the center of the tSNE plot (it is hard to tell from Figure 1—figure supplement 3C) or are these cells just biologically less distinct or require larger cell numbers to have sufficient coverage?

We generated some new figures regarding cells in the center being less tightly clustered. We observed that groups near the center may be less distinct from each other as seen by the max observed fold change for cluster markers and also the number of cluster markers (Figure 1—figure supplement 6B), but they don’t have lower coverage (as seen by Figure 1—figure supplement 6A).

We apologize for any confusion with unc-4. We have rewritten this in the main text to read, “For instance, unc-4 is currently the sole established marker for many cholinergic hemilineages, including 7B, 12A, 18B and 19B. We observe multiple cholinergic clusters enriched for unc-4 (Figure 1—source data 1; Figure 4—figure supplement 1B). To assign these clusters to known hemilineages, additional cluster specific enriched genes can be investigated by immunohistochemistry.”

[Editors' note: further revisions were suggested prior to acceptance, as described below.]

The manuscript has been greatly improved. There is one remaining issue that should be addressed before publication:One reviewer was not convinced by Figure 5F that the cluster markers are segregated, as there seems to be co-labeled cell bodies across both clusters 23B and 8B. This is the only example where antibody staining or anatomical verification was used to confirm the accuracy of the cluster markers. I think it is worth quantifying the extent to which cells are co-labeled or adding labels to the images that make the result more clear.

Text that discusses Figure 5 has been rewritten. Figure 5 has been modified to include label for hemilineage 9. We have included a new supplementary figure, Figure 5—figure supplement 3, with single sections to reveal the co-localization. We have also included three videos, a summary table of cell counts, and modified the main text to include the new figures, tables and videos.

And:Figure 1—figure supplement 3B: the label wrong in A, should be "Res. 1"

Typo fixed.